# A conformational switch regulates the ubiquitin ligase HUWE1

Bodo Sander[1], Wenshan Xu[2,3], Martin Eilers[2,4], Nikita Popov[2,3], Sonja Lorenz[1]*

[1]Rudolf Virchow Center for Experimental Biomedicine, University of Würzburg, Würzburg, Germany; [2]Comprehensive Cancer Center Mainfranken, Würzburg, Germany; [3]Department of Radiation Oncology, University Hospital Würzburg, Würzburg, Germany; [4]Theodor-Boveri-Institute, Biocenter, University of Würzburg, Würzburg, Germany

**Abstract** The human ubiquitin ligase HUWE1 has key roles in tumorigenesis, yet it is unkown how its activity is regulated. We present the crystal structure of a C-terminal part of HUWE1, including the catalytic domain, and reveal an asymmetric auto-inhibited dimer. We show that HUWE1 dimerizes in solution and self-associates in cells, and that both occurs through the crystallographic dimer interface. We demonstrate that HUWE1 is inhibited in cells and that it can be activated by disruption of the dimer interface. We identify a conserved segment in HUWE1 that counteracts dimer formation by associating with the dimerization region intramolecularly. Our studies reveal, intriguingly, that the tumor suppressor p14ARF binds to this segment and may thus shift the conformational equilibrium of HUWE1 toward the inactive state. We propose a model, in which the activity of HUWE1 underlies conformational control in response to physiological cues—a mechanism that may be exploited for cancer therapy.

*For correspondence: sonja. lorenz@virchow.uni-wuerzburg.de

Competing interests: The authors declare that no competing interests exist.

## Introduction

The human ubiquitin ligase HUWE1 (also known as ARF-BP1, MULE, LASU1, or HECTH9) regulates the stability of diverse cellular substrates and, in consequence, numerous physiological processes, including DNA replication and damage repair, cell proliferation and differentiation, and apoptosis (*Scheffner and Kumar, 2014*; *Li et al., 2015*; *King et al., 2016*; *Chen et al., 2016*). HUWE1 is over-expressed in many tumors (*Confalonieri et al., 2009*; *Adhikary et al., 2005*; *Chen et al., 2005*; *Yoon et al., 2005*; *Myant et al., 2016*) and has been studied intensively in this context. Surprisingly, both pro-oncogenic and tumor-suppressor functions of HUWE1 have been documented in different tumor models. This is reflected in the range of HUWE1 substrates: HUWE1 can target oncoproteins, such as N-MYC, C-MYC (*Myant et al., 2016*; *Zhao et al., 2008*; *Inoue et al., 2013*; *Zhao et al., 2009*), and MCL1 (*Myant et al., 2016*; *Zhong et al., 2005*; *Warr et al., 2005*), but also the tumor suppressor protein p53 (*Chen et al., 2005*) and the co-factor of MYC, MIZ1 (*Inoue et al., 2013*; *Yang et al., 2010*; *Peter et al., 2014*) for degradation. Additionally, the functional role of HUWE1 may be context-dependent, since HUWE1 is capable of activating the transcription of either MYC/ MIZ1-induced or -repressed target genes in a cell-type-dependent manner (*Adhikary et al., 2005*; *Myant et al., 2016*; *Inoue et al., 2013*; *Peter et al., 2014*).

HUWE1 associates with and is inhibited by the tumor suppressor p14ARF ('Alternative Reading Frame' in the CDKN2A gene) (*Chen et al., 2005*), a key activator of p53-mediated cell cycle arrest and apoptosis (*Ozenne et al., 2010*). p14ARF is best known for its ability to inhibit the HDM2 ubiquitin ligase that targets p53 for degradation (*Sherr, 2001*). However, p19ARF, the mouse orthologue of human p14ARF, inhibits proliferation of *p53-* and *Mdm2*-deficient mouse embryo fibroblasts, formally demonstrating that ARF has growth-suppressive functions that are independent of both p53

and MDM2 (*Weber et al., 2000*). The nature of the p53-independent functions of p14ARF has been intensively debated (*Sherr, 2006*). One candidate mechanism is the inhibition of nucleolar function and ribosome assembly, which is mediated by associations of p14ARF with nucleophosmin and the SUMO-protease SENP3 (*Bertwistle et al., 2004*; *Itahana et al., 2003*; *Maggi et al., 2014*). The regulation of HUWE1 activity by p14ARF is implicated in both the p53-dependent and p53-independent functions of p14ARF (*Chen et al., 2005*). However, the structural underpinnings of how p14ARF inhibits HUWE1 have remained elusive.

HUWE1 belongs to the HECT (<u>H</u>omologous to <u>E</u>6AP <u>C-T</u>erminus)-family of ubiquitin ligases (E3 enzymes) (*Huibregtse et al., 1995*). These enzymes are characterized by a conserved C-terminal catalytic domain, the 'HECT domain', that serves to transfer ubiquitin molecules from ubiquitin-loaded E2 enzymes onto substrate proteins (*Huang et al., 1999*). In an intermediate step, a catalytic cysteine residue of the HECT ligase forms a thioester bond with the C-terminus of ubiquitin. The activated carbonyl group of ubiquitin is then nucleophilically attacked by the ε-amino group of a lysine residue of the substrate. This results in the formation of an isopeptide linkage between ubiquitin and the substrate. If ubiquitin, in turn, acts as a substrate, this process leads to the formation of a ubiquitin chain.

The HECT domain consists of two lobes, a smaller C-terminal 'C'-lobe and a larger 'N'-lobe that are connected by a flexible linker (*Huang et al., 1999*; *Verdecia et al., 2003*). The N-lobe associates with E2 enzymes (*Huang et al., 1999*) and, at least in the NEDD4-subfamily, with a ubiquitin molecule of regulatory function (*Maspero et al., 2011*, *2013*; *Ogunjimi et al., 2010*; *French et al., 2009*; *Zhang et al., 2016*; *Kim et al., 2011*). In contrast, the C-lobe harbors the catalytic cysteine and interacts with the ubiquitin molecule that ought to be transferred to the substrate (*Maspero et al., 2013*; *Kamadurai et al., 2009*, *2013*; *Kim and Huibregtse, 2009*). Substrate proteins are typically recognized by regions outside of the HECT domain and oriented toward the catalytic center in ways that we are only beginning to understand (*Kamadurai et al., 2013*).

HUWE1 is a large protein of 482 kDa that is mostly uncharacterized structurally, with the exception of the C-terminal HECT domain (defined here as residues 3993–4374; PDB IDs: 3H1D (*Pandya et al., 2010*) and 3G1N), a ubiquitin-associated (UBA) domain (residues 1316–1355, PDB ID: 2EKK), a BH3 domain (residues 1976–1990, PDB ID: 5C6H) that interacts with MCL1 (*Zhong et al., 2005*; *Warr et al., 2005*), and a UBM1 domain of unknown function (residues 2951–3003, PDB ID: 2MUL). How these domains are arranged to mediate the versatile functions of HUWE1 in the cell has not been studied. It is also unknown whether HUWE1 contains ordered structural elements flanking the HECT domain that affect its catalytic functions, as observed for other members of the HECT E3 family (*Kamadurai et al., 2013*; *Muñoz-Escobar et al., 2015*; *Riling et al., 2015*; *Gallagher et al., 2006*).

Here, we report the surprising discovery that the activity of HUWE1 underlies conformational control through an intricate balance of dimerization and competing intra- and intermolecular interactions. Intriguingly, our studies suggest that p14ARF exploits this inherent conformational equilibrium of HUWE1 and may suppress HUWE1 activity by stabilizing its auto-inhibited state.

## Results

### The crystal structure of a C-terminal region of HUWE1 reveals an asymmetric dimer

To map structurally ordered regions of HUWE1 for crystallographic studies, we performed limited proteolysis of an extended C-terminal HUWE1 fragment (residues 3759 to 4374) (data not shown). Along with secondary structure predictions, these analyses led to the identification of a stable HUWE1 construct (residues 3951–4374) comprising the catalytic HECT domain and a previously uncharacterized 42-residue extension that readily crystallized. We determined the crystal structure of this construct at 2.7 Å resolution by molecular replacement, using a structure of the HUWE1 HECT domain (PDB ID: 3H1D [*Pandya et al., 2010*]) as a search model. The data collection and refinement statistics are summarized in *Table 1*; a comprehensive B-factor analysis is provided in *Figure 1—figure supplement 1*.

The structure has two molecules of HUWE1 in the asymmetric unit. The HECT domain adopts an overall similar structure in both molecules (backbone-RMSD ~0.5 Å), in which the HECT C-lobe

**Table 1.** X-ray crystallographic data collection and refinement statistics for the structure of HUWE1 (3951–4374). Values in parentheses correspond to the highest resolution shell. For a detailed B-factor analysis, see *Figure 1—figure supplement 1*.

| Data collection | |
|---|---|
| Wavelength (Å) | 0.9677 |
| Space group | P $6_3$ |
| Cell dimensions | |
| $a,b,c$ (Å) | 177.46 177.46 106.26 |
| $\alpha, \beta, \gamma$ (°) | 90.00 90.00 120.00 |
| Resolution (Å) | 46.15–2.70 |
| $R_{merge}$ | 0.058 (0.349) |
| $I/\sigma(I)$ | 16.2 (2.6) |
| $CC_{1/2}$ | 0.998 (0.828) |
| Completeness (%) | 99.4 (98.5) |
| Redundancy | 4.2 (2.9) |
| Wilson B factor | 51.1 |
| **Refinement** | |
| Resolution | 46.15–2.70 |
| Reflections used | 52025 |
| $R_{free}$ reflections | 2549 |
| $R_{work}/R_{free}$ | 0.197/0.227 |
| No. of non-hydrogen atoms | 7015 |
| Protein | 6983 |
| Ligands | 32 |
| Average $B$ factors | 70.84 |
| Protein | 70.97 |
| Solvent | 44.19 |
| RMSD from ideality | |
| Bonds (Å) | 0.005 |
| Angles (°) | 0.845 |
| Ramachandran statistics | |
| Favored (%) | 97.97% |
| Disallowed (%) | 0.00% |
| MolProbity clash score | 1.89 |

occupies a centric position along the N-lobe (*Figure 1A,B*). Such a conformation has been referred to as 'T'-shape (*Verdecia et al., 2003*; *Lorenz et al., 2013*) and is also seen in structures of the isolated HECT domain of HUWE1 (PDB IDs: 3H1D [*Pandya et al., 2010*] and 3G1N).

However, the conformation of the N-terminal extension differs between the two molecules in the asymmetric unit (*Figure 1B*). In molecule A, the C-terminal part of the extension (residues 3979–3990) that directly flanks the HECT domain, folds into an α-helix that we will refer to as the 'thumb helix'. In contrast, the same region adopts a turn followed by short $3_{10}$-helical segments in molecule B. The N-terminal part of the extension consists of a long amphipathic α-helix spanning 21 (3951–3971) and 22 (3951–3972) residues in molecules A and B, respectively, that we will term the 'pointer helix'. Interestingly, the pointer helix of molecule A points away from the HECT domain and contacts molecule B, where its hydrophobic helix face is grasped tightly by the thumb, the pointer helix, and part of the C-lobe of molecule B (*Figure 1C*).

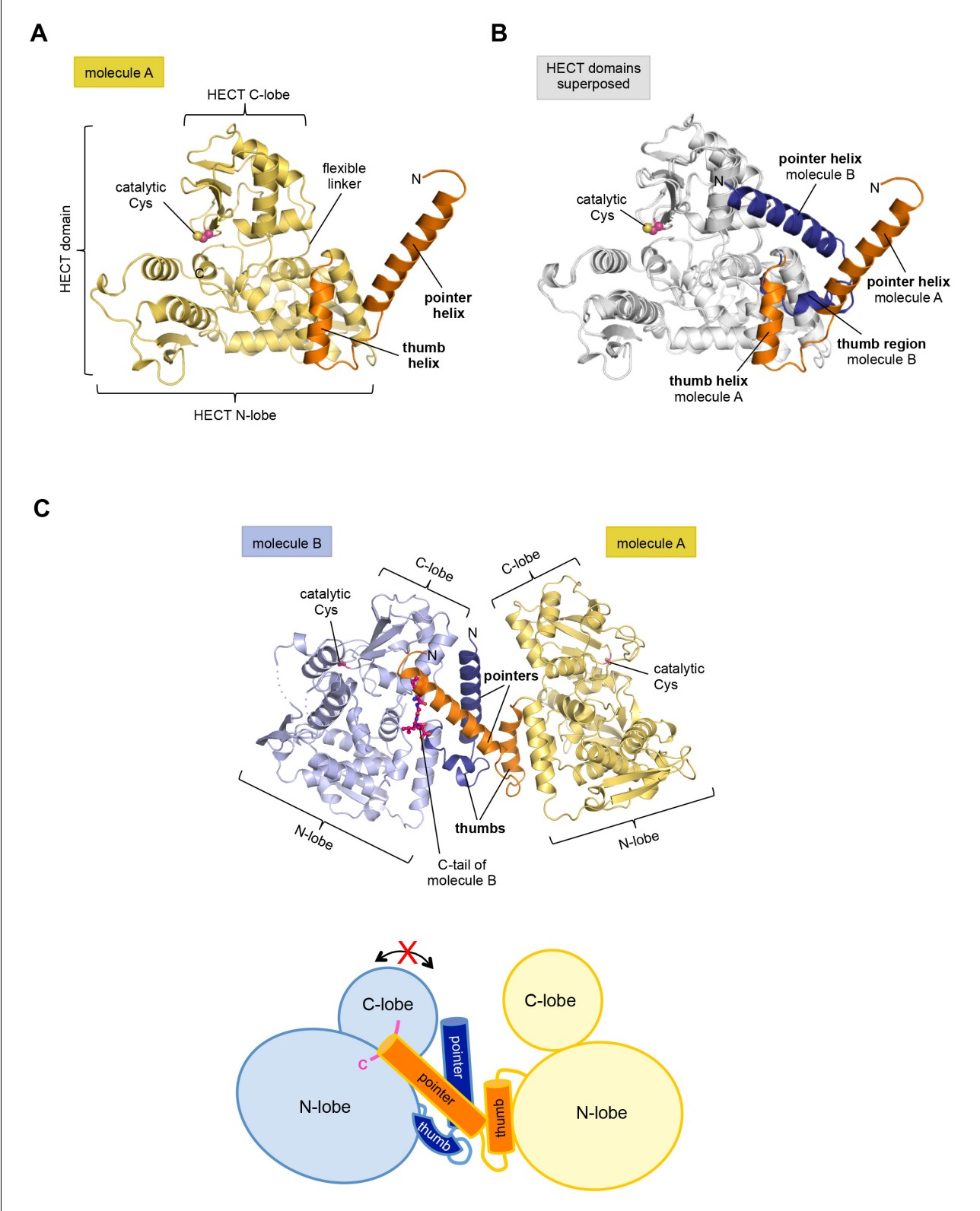

**Figure 1.** The crystal structure of the C-terminal region of HUWE1 reveals an asymmetric dimer. (**A**) Structural organization of HUWE1 (3951–4374), shown for molecule A. The bi-lobal catalytic HECT domain is colored yellow, the previously uncharacterized region orange. (**B**) Superposition of the HECT domains of molecules A and B (grey). The N-terminal regions are highlighted. (**C**) Asymmetric dimer formed of molecules A and B in the crystal. The C-terminal residues (4370–4374, 'C-tail') of molecule B are highlighted in magenta. Two loop regions in molecule B, distant from the dimer

*Figure 1 continued on next page*

Figure 1 continued

interface, are disordered (dotted lines) (top). Cartoon model of the asymmetric dimer (bottom). The C-lobe and the C-tail (magenta) of molecule B are in a locked conformation (see *Figure 2C,D*). In all structural figures, the protein backbones are rendered as cartoons and the side chains of relevant residues as ball-and-stick models.

The following figure supplement is available for figure 1:

**Figure supplement 1.** Crystallographic B-factor analysis.

The resulting asymmetric dimer interface buries a solvent-accessible surface area of ~1350 Å$^2$ of molecule A and ~1340 Å$^2$ of molecule B. Interface scoring programs, such as the 'Protein interfaces, surfaces and assemblies' service PISA at the European Bioinformatics Institute (http://www.ebi.ac.uk/pdbe/prot_int/pistart.html) (*Krissinel and Henrick, 2007*) predict a significant solvation free-energy gain upon dimer formation of −26 kcal/mol, indicating that the underlying contacts are extraordinarily hydrophobic. As illustrated by an open-book view of the dimer interface and a contact schematic (*Figure 2A,B*), the extensive hydrophobic contact network stretches all along the thumb and pointer helices of both molecules and includes Phe 3955, Phe 3958, Ala 3959, His 3962, Val 3965, Leu 3966, Ile 3969, Val 3984, Leu 3985, Tyr 3988, and Val 3991. Additional contacts are subunit-specific, due to the asymmetric nature of the dimer. For instance, the thumb region of molecule B, which is re-organized structurally compared to molecule A, is tilted outwards with respect to the HECT domain, thus allowing it to cradle the C-terminal part of the pointer helix of molecule A (*Figure 1C*). This configuration brings Phe 3982 of molecule B into a central position at the interface (*Figure 2A,B*).

Remarkably, the C-lobe of molecule B is conformationally locked at the dimer interface, where it buttresses the N-terminal part of the pointer helix of molecule A (*Figures 1C* and *2A,B*). C-lobe mobility, however, as enabled by a flexible linker between the N- and C-lobes of the HECT domain (*Figure 1A*), is required for HECT E3 activity (*Verdecia et al., 2003*). We thus predict that the conformation of molecule B in the crystal reflects an auto-inhibited state. This interpretation is underscored by the fact that the dimer interface buries a particular region of the C-lobe, including Leu 4335, Met 4359, and Leu 4362 (*Figure 2A*). The homologous region in NEDD4-type E3 enzymes has been shown to interact with ubiquitin, thereby facilitating the transfer of ubiquitin from the E2 to the E3 (*Maspero et al., 2013*; *Kamadurai et al., 2009*, *2013*) (*Figure 2—figure supplement 1A*). If HUWE1 oriented ubiquitin in a similar manner, this complex would clash with the HUWE1 dimer (*Figure 2—figure supplement 1B*).

Another indication that the conformation of molecule B is auto-inhibited comes from the positioning of the C-terminal region ('C-tail') that participates in the dimer interface (*Figures 1C* and *2*). This region contains a phenylalanine four residues from the C-terminus (Phe 4371), known as '−4 Phe', that is conserved across the HECT E3 family (*Salvat et al., 2004*) and has a critical role in the transfer of ubiquitin from its thioester-linked state at the E3 active site to lysine residues, presumably by anchoring the C-lobe at a specific position on the N-lobe (*Kamadurai et al., 2013*; *Salvat et al., 2004*) and/or by interacting with ubiquitin. Consistently, truncation of the C-lobe by four residues or mutation of −4 Phe was found to inhibit the auto-ubiquitination acitivites of various HECT E3 enzymes, including the HECT domain of HUWE1 (*Verdecia et al., 2003*; *Maspero et al., 2013*; *Kamadurai et al., 2013*; *Pandya et al., 2010*; *Salvat et al., 2004*; *Mari et al., 2014*). However, the position of −4 Phe has rarely been resolved in crystal structures of HECT domains (*Zhang et al., 2016*; *Kamadurai et al., 2013*), which may be due to the flexibility of the C-tail in the absence of substrates. Our crystal structure of HUWE1 has the C-tail of molecule B fully resolved (*Figure 2C*). −4 Phe is buried at the dimer interface, where it interacts with Phe 3955, Phe 3958, and the aliphatic side chain portion of Lys 3954 of molecule A in trans, and with Leu 4335 and Leu 4362 of the C-lobe in cis (*Figure 2D*). This tight engagement of −4 Phe at the dimer interface is incompatible with the C-tail reaching out to the N-lobe, as seen in an active conformation of the NEDD4-type E3 RSP5 (*Kamadurai et al., 2013*). Taken together, we conclude that the conformation of molecule B in our crystal structure of HUWE1 represents an auto-inhibited state.

In contrast to the locked C-lobe conformation of molecule B, the C-lobe of molecule A does not contact the dimer interface (*Figure 2A*) and its C-tail could not be modeled (*Figure 2—figure*

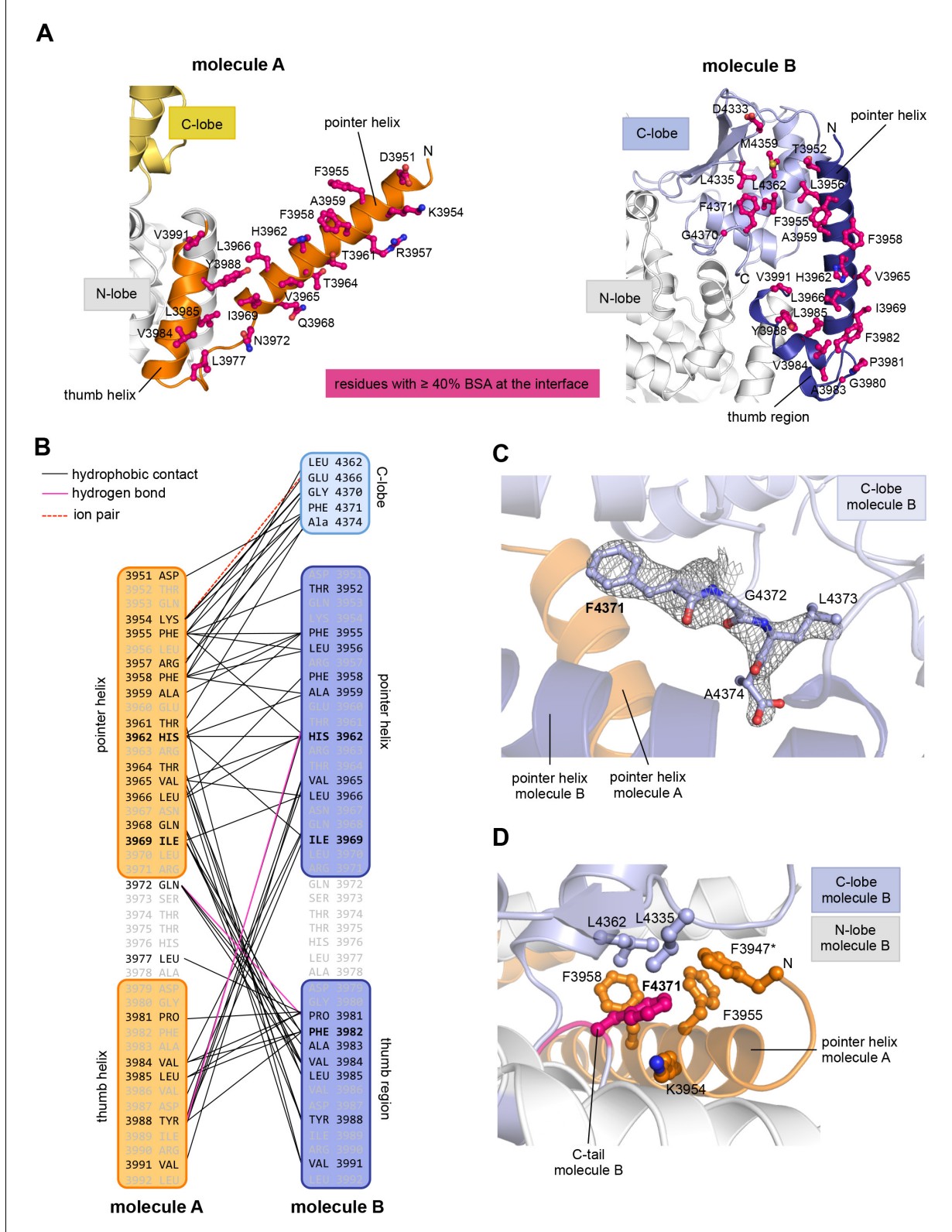

**Figure 2.** The dimer interface in HUWE1 is extraordinarily hydrophobic and locks the C-lobe of one subunit conformationally. (**A**) Open-book-view of the dimerization sites on molecules A and B of HUWE1 (3951–4374). The side chains of interfacing residues with ≥ 40% buried surface area (BSA) are displayed. (**B**) Schematic representation of the contact network between molecules A and B. The most important hydrophobic interactions are illustrated. Ion pairs and hydrogen bonds are sparse. The mutation sites, His 3962, Ile 3969, and Phe 3982, are marked bold (see **Figure 4C–E**).

*Figure 2 continued on next page*

*Figure 2 continued*

Residues that do not make contacts at the dimer interface are colored grey. For the C-lobe of molecule B, only those residues that form significant contacts and that have an interfacing BSA $\geq$ 40% are shown. (**C**) The C-tail of molecule B is buried at the dimer interface and well resolved in the electron density. An omit map countered at 1 $\sigma$ is shown around residues 4370–4374. Phe 4371 corresponds to the conserved '−4 Phe'. For the C-tail of molecule A, see *Figure 2—figure supplement 2*. (**D**) Detailed view of the hydrophobic environment of −4 Phe (magenta) of molecule B at the dimer interface. The asterisk marks 'F3947' as not being part of HUWE1, but of an N-terminal expression tag (see *Figure 3—figure supplement 2*).

The following figure supplements are available for figure 2:

**Figure supplement 1.** The HUWE1 dimer buries a hydrophobic region that mediates ubiquitin recognition in E3 enzymes of the NEDD4-subfamily.

**Figure supplement 2.** The C-tail of molecule A is disordered in the crystal structure.

**Figure supplement 3.** The position of the C-lobe of molecule A in the crystal is influenced by peripheral lattice contacts.

**Figure supplement 4.** The asymmetric HUWE1 dimer is compatible, in principle, with the C-lobe of molecule A adopting catalytically competent orientations, as seen for E3 enzymes of the NEDD4-subfamily.

*supplement 2*). Instead, the C-lobe of molecule A forms peripheral, predominantly electrostatic lattice contacts with three neighboring molecules in the crystal (*Figure 2—figure supplement 3*). Neither of these interfaces or putative higher order oligomeric assemblies are relevant in solution (see below, *Figures 3* and *4*). In principle, it is conceivable structurally that the C-lobe of molecule A is flexible in the context of the dimer and that it occupies catalytically relevant positions, as observed in other HECT E3 enzymes, in solution (*Figure 2—figure supplement 4*)

Taken together, our crystallographic studies reveal an asymmetric dimer of HUWE1 (3951–4374), in which one subunit adopts an auto-inhibited conformation.

## The C-terminal region of HUWE1 dimerizes in solution

To investigate if the crystallized construct, HUWE1 (3951–4374), dimerizes in solution, we performed size exclusion chromatography-coupled multi-angle light scattering (SEC-MALS) experiments at different protein concentrations, ranging from 3 to 378 µM at the stage of injection (*Figure 3A*). We find that HUWE1 elutes at smaller volumes with increasing protein concentrations, indicative of an increase in its hydrodynamic radius. At the lowest concentration tested, the MALS-derived molecular weight (MW) of 54 kDa is very close to the MW of a monomer (53 kDa). With increasing protein concentration the conformational equilibrium shifts toward a dimer. At the highest concentration tested, the MALS analysis yields a MW of 93 kDa, in line with a dominant population of the dimeric state.

We also used small-angle X-ray scattering (SAXS) to study the oligomerization state of HUWE1 (3951–4374) in solution. *Figure 3B* shows the scattering profile measured for a sample of ~40 µM concentration, at which the protein is expected to be predominantly dimeric (note that the sample is not diluted during SAXS, in contrast to SEC-MALS experiments). Indeed, simulated scattering profiles based on the crystal structure of the HUWE1 dimer approximate the experimental data well (*Figure 3B*), as reflected by an excellent fit score, $\chi$, of 1.4 for a single model (average fit score for the single best models of three replicate runs: $\chi = 1.4 \pm 0.1$). In contrast, simulations based on the structures of either of the monomers seen in the crystal are unable to reproduce the experimental data well, with $\chi$−values worsening to 19.9 and 17.7 for molecules A and B, respectively (*Figure 3—figure supplement 1*; average fit scores for the single best models of three replicate runs: $\chi = 20.3 \pm 0.1$ and $17.6 \pm 0.2$, respectively). Assuming that the dimerization region is unstructured in the context of a monomer results in similarly insufficient fit scores, even when applying multiple ensemble search approaches (data not shown; for details, see Materials and methods). The SAXS data are, therefore, inconsistent with a monomeric state of HUWE1 (3951–4374), but they are in agreement with the structure of the crystallographic dimer.

In line with these analyses, the SAXS-derived radius of gyration ($R_g$), a measure of the molecular mass distribution, of HUWE1 (3951–4374) is significantly larger than the value expected for a monomer (experimental $R_g$ = 36 Å; calculated $R_g$ = 25 Å) and is, instead, consistent with the value

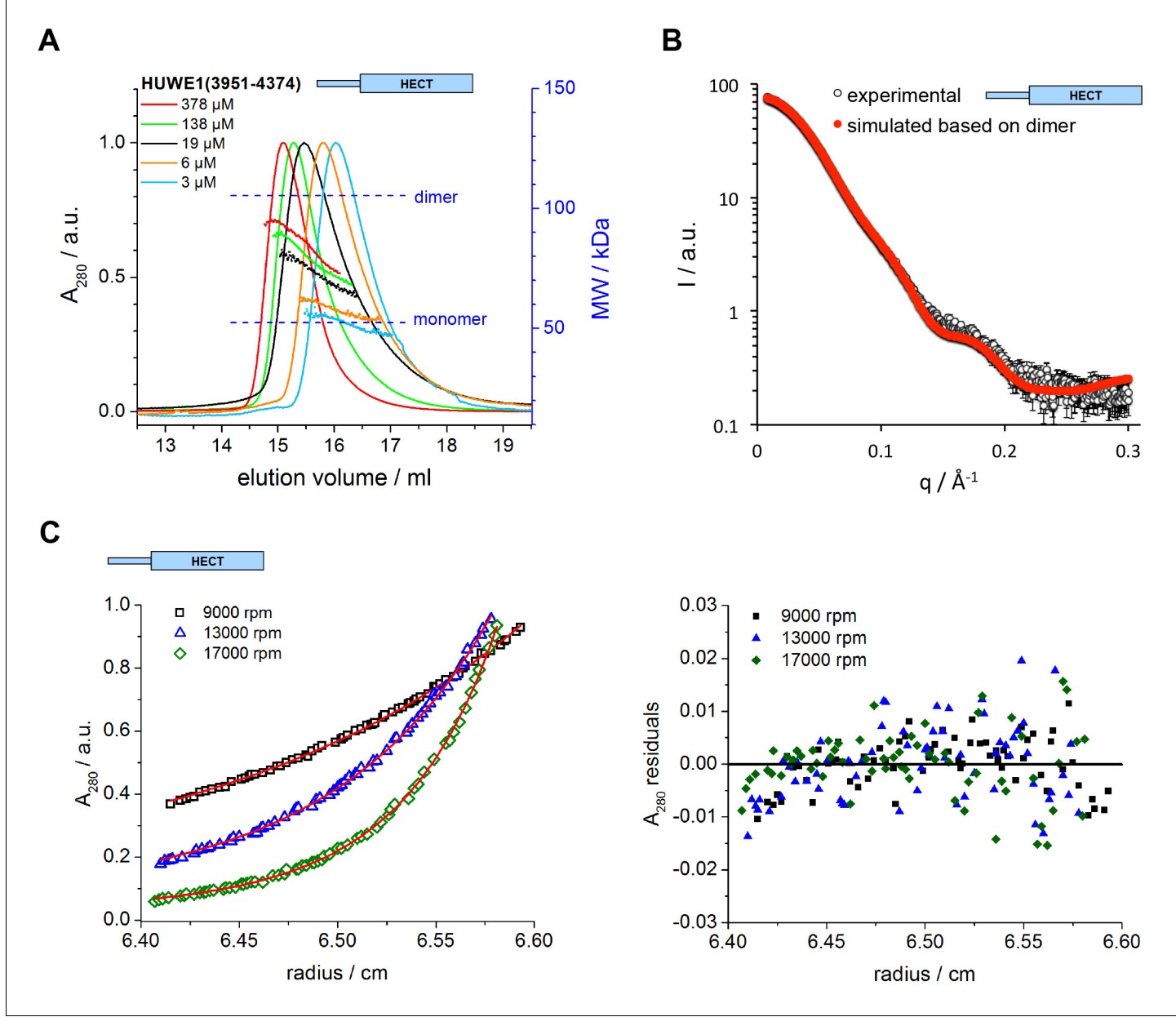

**Figure 3.** The C-terminal region of HUWE1 dimerizes in solution. (**A**) SEC-MALS experiments with the crystallization construct, HUWE1 (3951–4374), at different concentrations. The heights of the absorbance peaks were normalized to a value of 1. The MALS-based MW profiles for each elution peak are shown, along with the calculated MWs of the monomer (53 kDa) and dimer (106 kDa), as references. The MALS-derived MWs in the order of increasing protein concentration are 54, 60, 76, 90, and 93 kDa. (**B**) SAXS experiment with HUWE1 (3951–4374) at 40 µM concentration. The experimental SAXS intensity, I, plotted over the momentum transfer, q, is compared to a simulated scattering curve, based on the crystal structure of the HUWE1 dimer. The corresponding fit score, $\chi$, is 1.4. Analogous simulations of scattering curves, based on the structures of the two HUWE1 monomers seen in the crystal are shown in *Figure 3—figure supplement 1*. (**C**) AUC sedimentation equilibrium experiments with HUWE1 (3951–4374) at 10 µM concentration. The protein distribution across the cell, monitored by absorbance ($\lambda$ = 280 nm), is shown for three rotation speeds (left). The data were fitted with Sedphat (*Houtman et al., 2007*) (red line), yielding a $K_D$ of 2.9 ± 1.5 µM. The residuals of the fit are shown (right).

The following figure supplements are available for figure 3:

**Figure supplement 1.** The SAXS data for HUWE1 (3951-4374) are inconsistent with a monomeric state.

**Figure supplement 2.** The N-terminal expression tag that was present in the crystallized construct does not induce the dimerization of HUWE1 in solution.

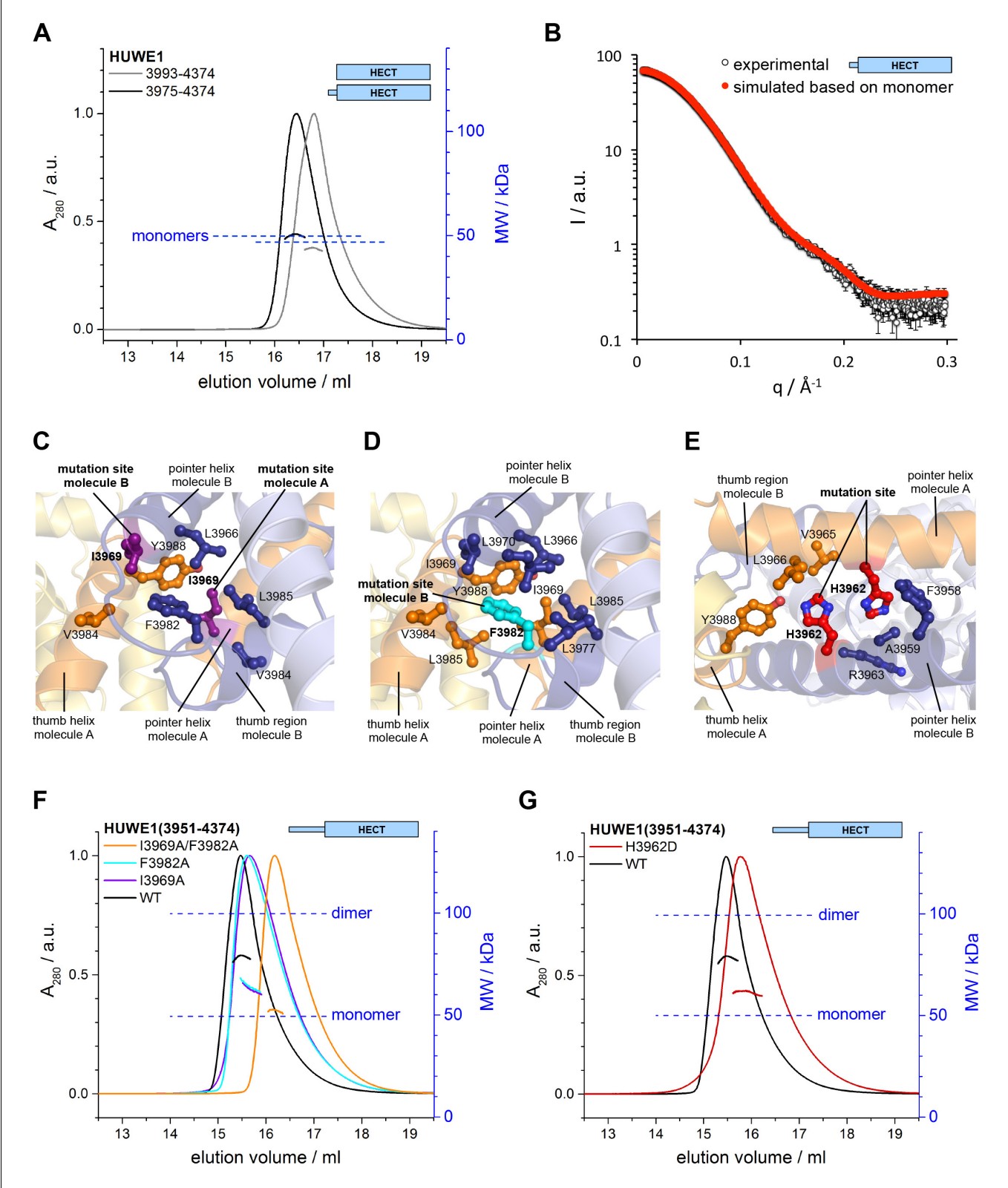

**Figure 4.** Key contacts in the crystallographic dimer interface mediate the dimerization of HUWE1 in solution. (**A**) SEC-MALS experiments with a truncated construct, HUWE1 (3975–4374), that lacks the pointer helix and with the HECT domain (HUWE1 [3993–4374]), at 380 and 400 μM, respectively. The absorbance peak heights were normalized to a value of 1. The MALS-derived MWs are 51 kDa and 44 kDa; the calculated MWs for the monomers are 50 and 47 kDa, respectively. (**B**) Comparison of the experimental SAXS intensity profile, I(q), of HUWE1 (3975–4374) at 40 μM concentration with a

*Figure 4 continued on next page*

*Figure 4 continued*

simulated scattering curve, based on the crystal structure of a monomer, as extracted from the crystal structure of HUWE1 (3951–4374). The fit score, χ, is 1.3. (C–E) Detailed views of the hydrophobic side chain environments (within a radius of ~4 Å) of the mutation sites Ile 3969 (purple) of molecules A and B (C), Phe 3982 (cyan) of molecule B (D), and His 3962 (red) of molecules A and B (E). For the environment of Phe 3982 of molecule A, see *Figure 4—figure supplement 1*. (F) SEC-MALS experiments with WT and mutated variants of HUWE1 (3951–4374) at ~350 µM concentration. The MALS-derived MWs of I3969A, F3982A, and I3969A/F3982A are 63, 64, and 53 kDa, respectively, compared to 79 kDa for the WT. The calculated MW of each protein monomer is 50 kDa. (G) Analogous SEC-MALS experiment with the tumor-associated mutated variant H3962D of HUWE1 (3951–4374). The MALS-derived MW is 62 kDa; the calculated MW of a monomer is 50 kDa. For comparison, the WT profile is displayed (same as in panel F).

The following figure supplement is available for figure 4:

**Figure supplement 1.** In the asymmetric dimer Phe 3982 of molecule A engages in intramolecular interactions.

---

calculated for the crystallographic dimer ($R_g$ = 35 Å). Taken together, these data demonstrate that HUWE1 (3951–4374) dimerizes in solution.

As noted above, our crystallization construct of HUWE1 contained an N-terminal TEV protease-cleavable expression tag. A small portion of this tag (residues 3947*−3950* of molecule A, residue 3950* of molecule B, and a short, unconnected stretch of residues 3936* to 3941* that can not be assigned to either chain unambiguously, due to missing electron density) is visible in the crystal structure and located near the dimer interface (*Figure 3—figure supplement 2A*). Unlike the dimerization region that is well defined in the electron density, the residues of the tag have rather poor electron density and high B-factors (*Figure 1—figure supplement 1*). Also, hydrophobic contacts between residues of the tag and HUWE1 are few compared to the extensive interface formed by the dimerization region (*Figure 3—figure supplement 2A*). However, these contacts include one residue, Phe 3947*, in proximity of −4 Phe of molecule B (*Figure 2D*).

To ascertain that untagged HUWE1 lacking Phe 3947* still dimerizes, we compared the behavior of HUWE1 (3951–4374) in SEC-MALS experiments before and after cleavage of the tag (*Figure 3—figure supplement 2B*). We find that the MALS-derived MW difference between the untagged and tagged protein variants (7 kDa) is relatively close to the expected MW difference (~5.5 kDa), as calculated from the MW of the expression tag and upon assuming a constant fraction of dimer in both conditions. We can, therefore, conclude that the dimerization of HUWE1 in solution is not induced by the expression tag. Nevertheless, all in vitro functional analyses that are described below were performed with untagged constructs of HUWE1.

Importantly, we also demonstrate that untagged HUWE1 (3951–4374) dimerizes using analytical ultracentrifugation (*Figure 3C*). The corresponding dissociation constant, $K_D$, of ~3 µM, indicates that the dimerization of this construct is relatively weak in vitro, consistent with our SEC-MALS analyses.

## Key contacts in the crystallographic dimer interface mediate the dimerization of HUWE1 in vitro and its self-association in cells

To test if the dimerization of HUWE1 in solution occurs through the same key contacts that make up the crystallographic dimer interface, we determined the effects of truncations and site-specific mutations on dimer formation. We generated an N-terminal truncation construct comprising residues 3975 to 4374 of HUWE1, in which the pointer helix is removed. Since the pointer helix is an integral part of the dimer interface, we expected the truncated construct to be monomeric. Indeed, this construct no longer dimerizes in SEC-MALS experiments, even at high concentrations of 380 µM, and its elution volume approaches that of the monomeric HECT domain (*Figure 4A*). The MALS-derived MWs of 51 and 44 kDa for HUWE1 (3975–4374) and for the HECT domain are close to the calculated MWs of 50 and 47 kDa, respectively.

In line with these observations, the experimental SAXS profile of HUWE1 (3975–4374) is approximated well by a simulated profile based on the monomeric structure of this fragment, as extracted from our crystal structure (single best fit score: χ = 1.3; average fit score for the single best models of three replicate runs: χ = 1.30 ± 0.05) (*Figure 4B*). The corresponding SAXS-derived $R_g$-value (27 Å) is also consistent with the calculated value for a monomer (26 Å).

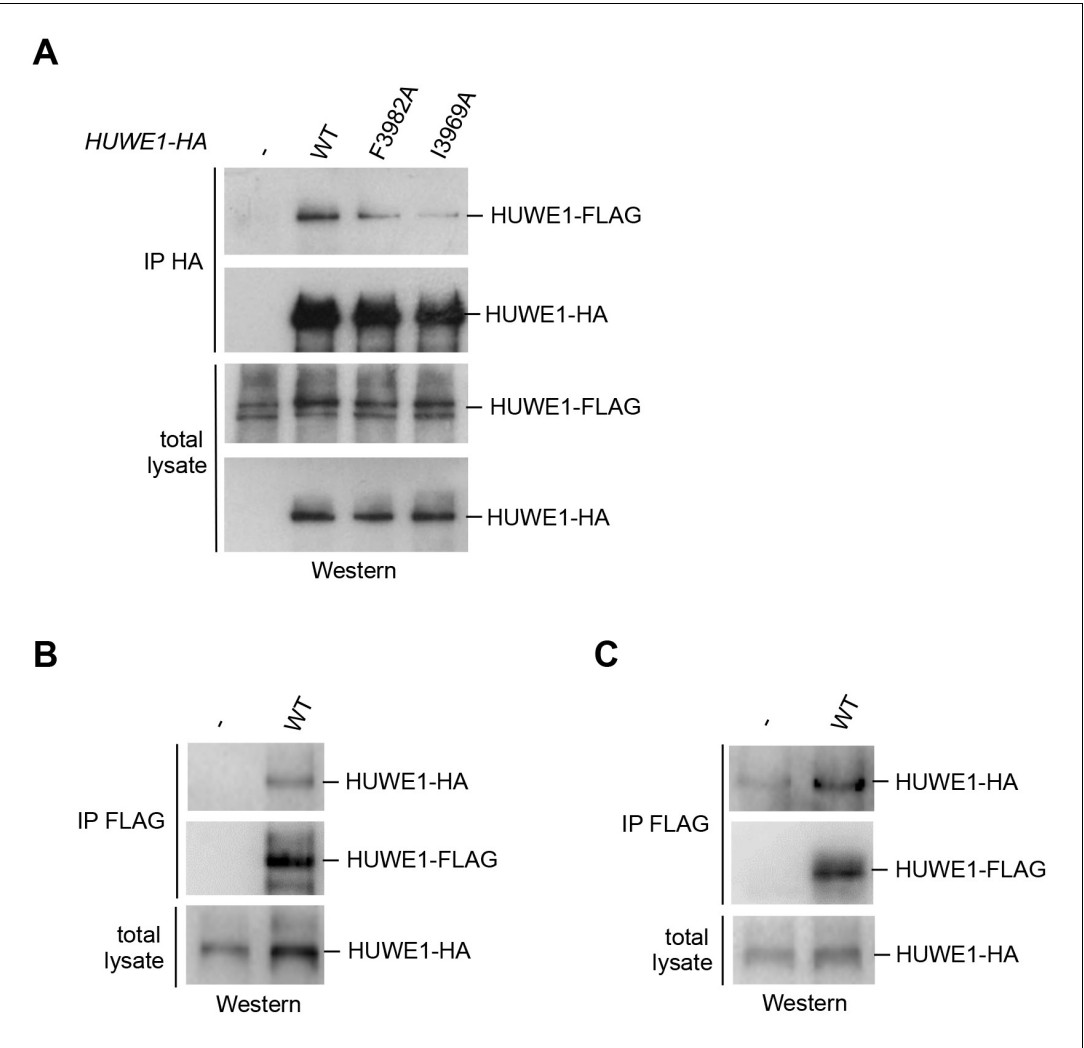

**Figure 5.** Self-association of HUWE1 occurs in cells and is mediated by the same key contacts as the dimerization observed in vitro. (**A**) Co-IP experiments from HeLa cells upon transient transfection with HA- and FLAG-tagged HUWE1 (2474–4374) WT, I3969A, and F3982A. (**B**) Co-IP experiments from HeLa cells stably expressing HA- and FLAG-tagged HUWE1 (2474–4374) WT that were generated by lentiviral transduction. (**C**) Co-IP experiments from HeLa cells stably expressing HA- and FLAG-tagged HUWE1 (2474–4374) WT that were generated by transposon-mediated gene delivery.
The following figure supplement is available for figure 5:

**Figure supplement 1.** Comparison of the HUWE1 expression levels in stable cell lines and upon transient transfection.

We next analyzed the effects of structure-based point mutations on the dimerization behavior of the crystallization construct, HUWE1 (3951–4374), in solution. We replaced Ile 3969, located near the C-terminus of the pointer helix, and Phe 3982, located near the N-terminus of the thumb helix, respectively, by alanine. Both residues form key hydrophobic contacts at the dimer interface: Ile 3969 of molecule A is at the core of the dimer interface, where it packs tightly against Leu 3966, Phe 3982, Val 3984, and Leu 3985 of subunit B (*Figure 4C*). In contrast, hydrophobic contacts of Ile 3969 in cis are rather peripheral and limited to Leu 3985 and Tyr 3988 of molecule A, due to a stark dislocation of the pointer helix from the body of molecule A (*Figures 4C,D* and *2A*). In molecule B Ile 3969 also points toward the dimer interface, where it contacts Val 3984 and Tyr 3988 of molecule A

(*Figure 4C*). In this case, Ile 3969 has an additional role in positioning Phe 3982 in cis, due to the 'compacted' arrangement of the pointer helix close to the body of molecule B (*Figures 4C* and *2A*).

Phe 3982 of molecule B, in turn, makes pivotal hydrophobic contacts with molecule A, involving Ile 3969, Val 3984, Leu 3985, and Tyr 3988 (*Figure 4D*). In addition, Phe 3982 is embedded into a hydrophobic network in cis that includes Leu 3966, Ile 3969, Leu 3970, Leu 3977, and Leu 3985. Phe 3982 of molecule A does not contribute to the dimer interface and instead interacts with the HECT N-lobe in cis (*Figure 4—figure supplement 1*). Taken together, both mutation sites, Ile 3969 and Phe 3982, are expected to contribute critically to the stability of the dimer interface.

We subjected the single mutant variants I3969A and F3982A of HUWE1 (3951–4374) to SEC-MALS experiments (*Figure 4F*). Both proteins elute at larger volumes than the WT. The MALS-derived MWs of 63 and 64 kDa for the I3969A and F3982A variants, respectively, are significantly lower than the MW derived for the WT (79 kDa), indicating that the conformational equilibria of these variants are shifted toward the monomeric state. The calculated monomeric MWs of all three constructs are 50 kDa. The simultaneous mutation of both sites, I3969A/F3982A, results in an additional increase in the SEC-elution volume (*Figure 4F*). The corresponding MALS-derived MW of 53 kDa is now close the calculated monomeric weight (50 kDa), which demonstrates that this protein variant does not dimerize significantly any more.

Notably, one of the mutation sites that we selected for our studies, Ile 3969, is among a subset of residues in the dimerization region that were found to be mutated in human tumors (cutaneous squamous cell carcinoma: I3969N and I3969V [*Pickering et al., 2014*]). Another cancer-associated, non-conservative amino acid substitution occurs at His 3962 (upper aerodigestive tract carcinoma: H3962D [http://cancer.sanger.ac.uk/cosmic/]). This residue adopts a central position at the hydrophobic face of the pointer helix and contributes to the dimer interface on both subunits by contacting Val 3965, Leu 3966, Tyr 3988 of molecule A and Phe 3958, Ala 3959, and the aliphatic portion of Arg 3963 of molecule B (*Figure 4E*). Consequently, we expected the H3962D variant of HUWE1 (3951–4374) to be impaired in dimer formation. Indeed, SEC-MALS experiments show that this variant elutes at larger volumes than the WT (*Figure 4G*), and the MALS-derived MW of 62 kDa confirms that the dimerization capacity of this variant is reduced significantly.

We next determined whether HUWE1 oligomerizes in the cellular context. To this end, we employed co-immunoprecipitation (co-IP) experiments from HeLa cells transiently transfected with FLAG- and HA-tagged versions of an extended HUWE1 construct spanning residues 2474 to 4374, as used in previous studies (*Adhikary et al., 2005*; *Peter et al., 2014*). Strikingly, we find that HUWE1 self-associates (*Figure 5A*). In contrast to WT HUWE1, the F3982A and I3969A variants both show a marked reduction in their abilities to associate with WT HUWE1, analogous to the disruptive effect of these mutations on HUWE1 dimerization that was observed in vitro (*Figure 5A*). The same results were obtained in inverse co-IP experiments (FLAG-IP; data not shown).

To assess the effect of the protein expression level on the self-association of WT HUWE1, we generated cell lines stably expressing HUWE1, using lentivirus- and transposon-mediated gene delivery, respectively. Notably, the cell line generated by lentiviral transduction shows higher HUWE1 levels than the transiently transfected cells (*Figure 5—figure supplement 1*) and, consistently, strong self-association of HUWE1 (*Figure 5B*). In contrast, the stable cell lines generated using transposons express HUWE1 at significantly lower levels than the transiently transfected cells (*Figure 5—figure supplement 1*). Yet, HUWE1 also self-associates robustly at these lower concentrations (*Figure 5C*).

Taken together, these studies confirm that HUWE1 dimerizes in vitro and self-associates in cells and that both processes depend on the same key contacts that mediate the formation of the crystallographic dimer.

## The disruption of the dimerization interface enhances the activity of HUWE1 in vitro and in cells

The fact that the dimerization of HUWE1 (3951–4374) is mediated by rather weak interactions suggests that it is a dynamic process in solution, in which the dimerization regions associate with each other and with the C-lobes of each molecule transiently. Considering that one subunit of the crystallographic dimer presents an auto-inhibited state, we thus expected the ensemble of molecules to be inhibited in solution and a disruption of the dimer interface to enhance the catalytic activity of HUWE1 overall. We tested this idea by comparing the auto-ubiquitination activities of HUWE1 (3951–4374) WT with the dimerization-deficient variants I3969A, F3982A, I3969A/F3982A, and

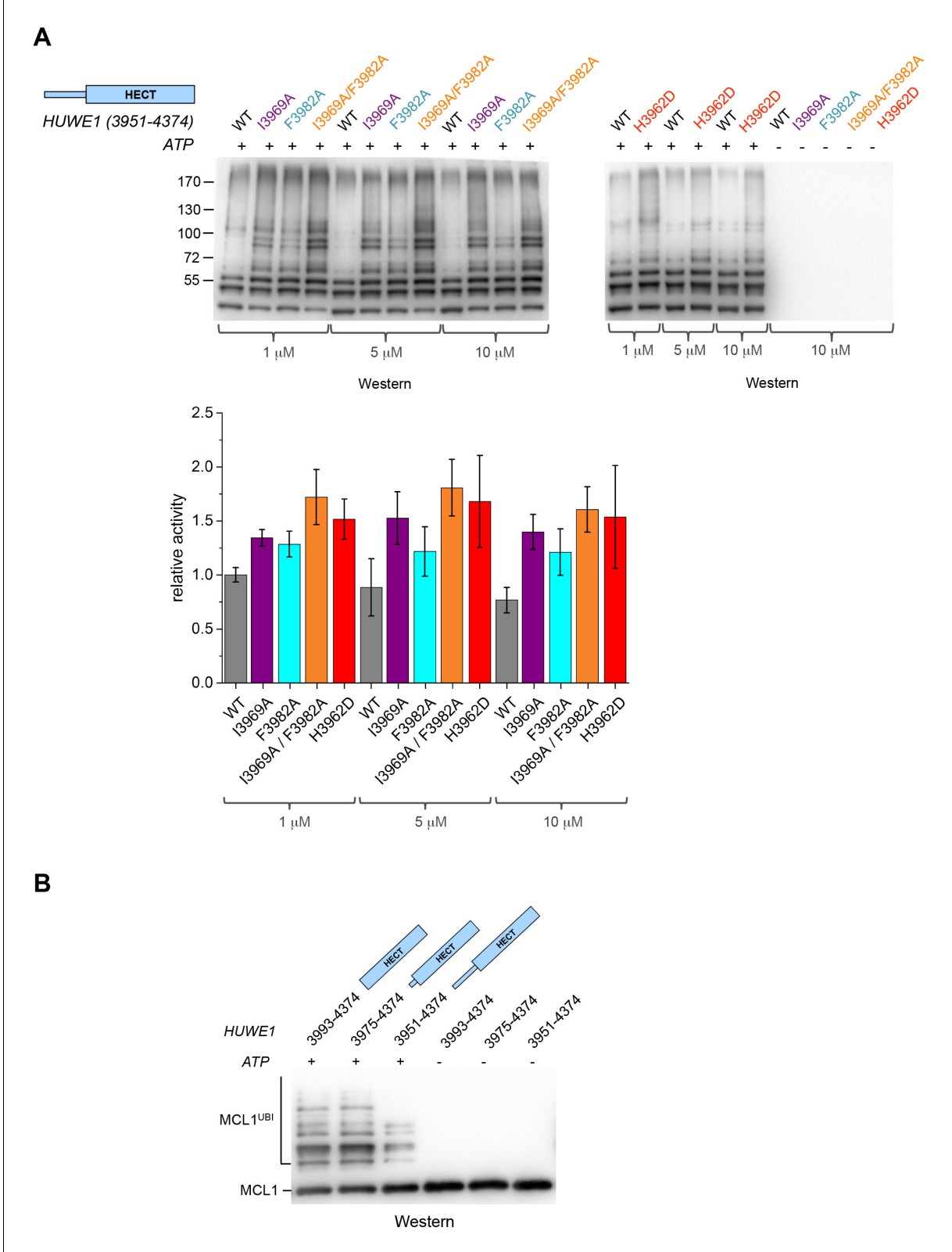

**Figure 6.** Disruption of the dimer interface enhances the activity of HUWE1 in vitro. (**A**) Comparison of the auto-ubiquitination activities of HUWE1 (3951–4374) WT to dimerization-deficient mutated variants at three different concentrations (1, 5, and 10 µM) in vitro, monitored by anti-ubiquitin Western blotting. The reactions contain HUWE1, E2 enzyme (UBCH7), E1 enzyme (UBA1), ubiquitin, and ATP/Mg$^{2+}$ (for details see Materials and methods). Negative controls were performed without ATP. The band intensities for HUWE1 species modified with at least four ubiquitin molecules were

*Figure 6 continued on next page*

Figure 6 continued

quantified with ImageJ (*Abràmoff and Magalhães, 2004*) and normalized by the intensity observed in the presence of the lowest concentration of HUWE1 WT. The error bars result from three independent reaction replicates. (B) Comparison of the ubiquitination activities of HUWE1 (3951–4374), HUWE1 (3975–4374) (pointer helix missing), and HUWE1 (3993–4374) (HECT domain only) at 5 µM concentration toward the substrate MCL1, monitored by anti-MCL1 Western blotting (for details, see Materials and methods).

H3962D in vitro at concentrations of 1, 5, and 10 µM, that is, below and above the $K_D$-value of dimerization (*Figure 6A*). We find that all of the mutated variants are, indeed, more active than the WT. The stimulatory effect is most pronounced for the double mutant variant, I3969A/F3982A, consistent with the dimer being most destabilized for this variant. The activity of HUWE1 WT decreases slightly with increasing protein concentrations, which can be explained, at least qualitatively, by concentration-dependent changes in the population of the dimeric state.

We also sought to compare the catalytic activity of HUWE1 (3951–4374) to the dimerization-deficient truncated construct, HUWE1 (3975–4374) and to the monomeric HECT domain (HUWE1 [3993–4374]). In this context, auto-ubiquitination assays are hard to interpret, since the individual constructs display different levels of auto-ubiquitination activities per se, due to their different numbers of ubiquitination sites and their distinct conformations. Therefore, we monitored the ubiquitination activities of these constructs toward a physiological substrate of HUWE1, MCL1 (*Figure 6B*). In line with our model, we find that MCL1 is ubiquitinated significantly more strongly by the truncated, dimerization-deficient constructs of HUWE1 than by HUWE1 (3951–4374) that dimerizes.

We next asked whether the dimerization of HUWE1 also affects its catalytic activity toward substrates in the cell. Interestingly, our studies show that both MCL1 and C-MYC are, indeed, ubiquitinated more strongly upon expression of the dimerization-deficient HUWE1 variants I3969A and F3982A than in the presence of the WT enzyme (*Figure 7A,B*). It should be noted, however, that the stimulatory effect toward either substrate is significantly more pronounced for the F3982A than for the I3969A variant. As a negative control, we included a catalytically dead variant of HUWE1, in which the active site cysteine is replaced by serine (C4341S). Interestingly, the low background level of ubiquitination observed in this case is similar to the one seen for the WT. Evidently, HUWE1 exists in an, at least partially, inhibited state in HeLa cells.

Taken together, these studies demonstrate that a disruption of the dimer interface of HUWE1 releases inhibitory restraints on its catalytic activity, both in vitro and in cells.

## The dimerization capacity of HUWE1 is modulated by intramolecular interactions

The observation that HUWE1 can adopt an auto-inhibited state made us wonder what mechanisms counteract dimerization for the enzyme to reach full activity. We thus set out to explore the role of an uncharacterized ~200 residue region N-terminal to the pointer helix. This region is predicted to contain four alternating stretches of α-helical content and low-sequence complexity, respectively (*Figure 8A*). Accordingly, we designed four additional HUWE1 constructs comprising residues 3896–4374, 3843–4374, 3810–4374, and 3759–4374. We subjected each of these constructs to SEC and SEC-MALS studies at protein concentrations, at which the crystallization construct, HUWE1 (3951–4374), exists predominantly as a dimer. A superposition of the SEC-profiles is shown in *Figure 8B*; the calculated and MALS-derived MWs are summarized in *Table 2*.

For HUWE1 (3896–4374), that is elongated by a 55-residue low complexity region compared to the crystallization construct and that has a calculated MW of 60 kDa, the MALS analysis yields a MW of 92 kDa. This elongated construct, therefore, also dimerizes. Surprisingly, however, an additional construct extension by a preceding α-helical region (HUWE1 [3843–4374]) results in a marked shift of the SEC elution peak toward larger volumes (*Figure 8B*) and a MALS-derived MW of 60 kDa that approximates the calculated monomeric MW of this construct (65 kDa). Similarly, the two longest constructs, HUWE1 (3810–4374) and (3759–4374), do not dimerize, as revealed by MALS analysis (*Table 2*). Interestingly, however, these two constructs elute at smaller volumes than the shorter constructs that dimerize. This elution behavior, as determined by the hydrodynamic radii of the proteins, reflects a rather extended shape of the larger, monomeric constructs compared to the shorter, dimeric ones.

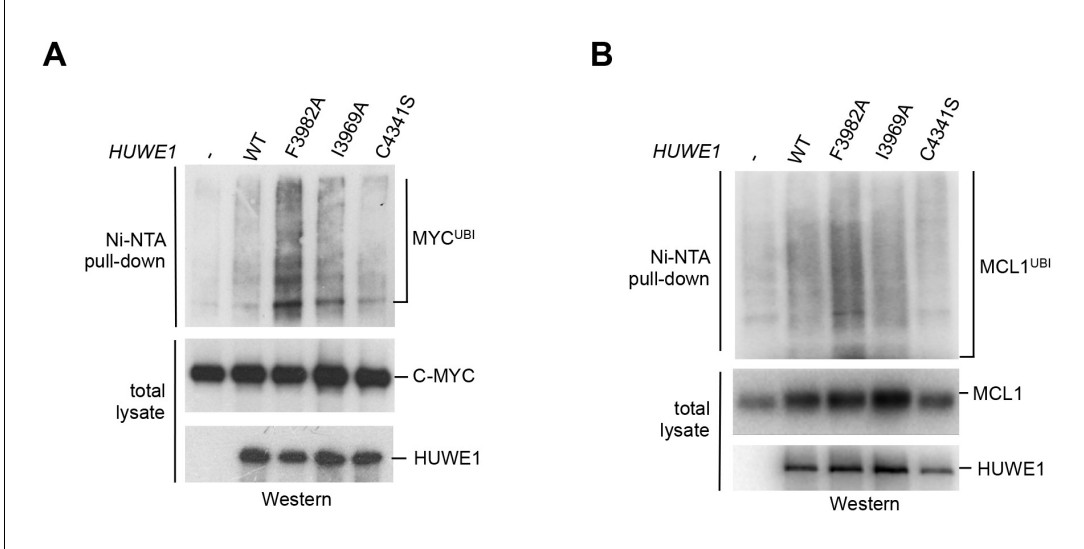

**Figure 7.** Mutations in the dimer interface enhance the activity of HUWE1 in cells. (**A,B**) Cell-based activity assays monitoring the substrate ubiquitination activities of HUWE1 (2474–4374) WT, the dimer interface mutants F3982A and I3969A, and the catalytically dead variant C4341S. HeLa cells were co-transfected with HUWE1 constructs, C-MYC, and MCL1, respectively. For details, see Materials and methods.

Taken together, these experiments reveal that the dimerization of HUWE1 through the thumb and pointer helices, as seen for HUWE1 (3951–4374), is unperturbed by the adjacent low complexity region (3896–3950), but that it is diminished in the presence of an additional segment of predicted α-helices, spanning residues 3843 to 3895.

To corroborate these findings we determined $R_g$-values for all HUWE1 constructs at ~40 μM concentration by SAXS (*Table 2*, *Figure 8—figure supplement 1*). In line with the SEC-MALS analysis, we find that those constructs that contain the dimerization region, but do not encompass residues 3843–3895 are characterized by relatively large $R_g$-values, as expected for predominantly dimeric populations. A pronounced reduction in the $R_g$-values, however, is observed upon including residues 3843–3895, indicating a shift in the conformational ensemble toward the monomeric state.

To test if the conformational switch in HUWE1 that is triggered by the presence of residues 3843-3895 coincides with a stimulation of activity, we monitored the ubiquitination activities of different HUWE1 constructs toward MCL1 (*Figure 8C*). Indeed, HUWE1 (3843–4374) displays similarly high activity as the isolated HECT domain. In comparison, the N-terminally truncated constructs that do not contain residues 3843–3895 and that dimerize (HUWE1 [3896–4374] and [3951–4374]) are significantly less active. We we will thus refer to the critical monomer-inducing region of HUWE1, residues 3843-3895, as the 'activation segment'.

In principle, the activation segment may antagonize the dimerization of HUWE1 either through direct intramolecular interactions with the dimerization region or through allosteric effects. When comparing the amino acid sequences of the activation segment and the dimerization region, we noticed a striking degree of similarity (*Figure 9A*). In particular, a hydrophobic motif in the thumb region, 'FAVLVxxxxV' (residues 3982–3991) recurs in almost identical form in the activation segment (residues 3874–3883). Remarkably, most of the residues in the dimerization region that are conserved in the activation segment (or that are at least substituted by similar amino acids) coincide with residues that participate in the dimer interface. Those include Ile 3969 and Phe 3982 that we found to be critical for dimer stability. Based on these analyses, we hypothesized that the activation segment may be able to interact with the dimerization region intramolecularly. This scenario is plausible structurally, since the 55-residue linker connecting the two regions is predicted to be disordered.

To test whether the activation segment and the dimerization region bind to each other, we conducted SEC experiments with the activation segment (residues 3843–3902 of HUWE1, fused N-terminally to maltose-binding protein [MBP]), HUWE1 constructs lacking the activation segment, and

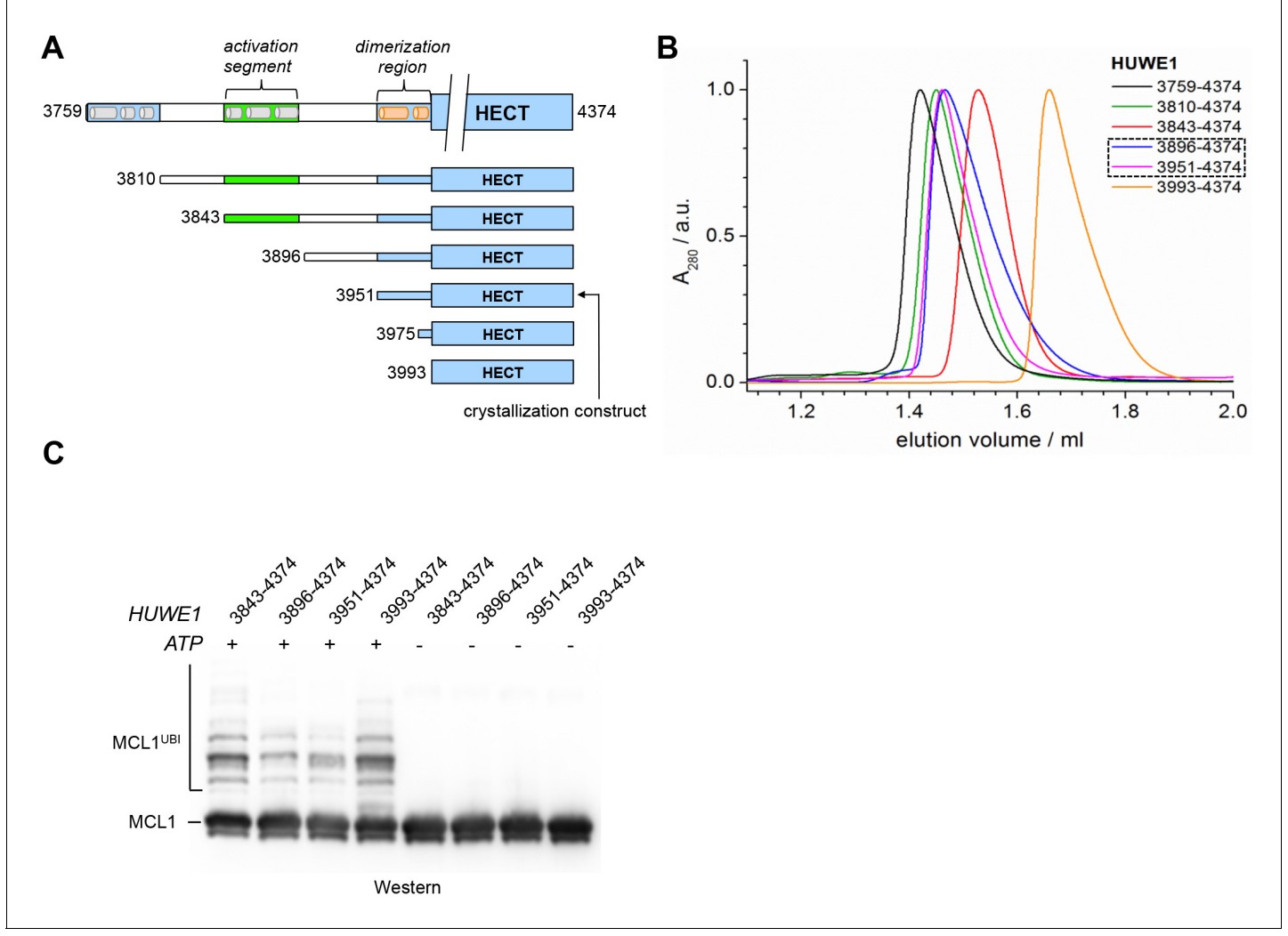

**Figure 8.** A segment located N-terminal to the dimerization region interferes with the dimerization of HUWE1 and stimulates its activity. (**A**) The panel of HUWE1 constructs used in our in vitro studies. Helices >5 residues, as predicted by I-TASSER (*Yang et al., 2015*) in the structurally uncharacterized regions are colored grey. White boxes mark regions of low sequence complexity. (**B**) SEC experiments with six individual HUWE1 constructs at 120 µM concentration. The absorbance peak heights were normalized to a value of 1. HUWE1 (3896–4374) and (3951–4374) that dimerize are marked by a dashed box. For the calculated and MALS-derived MWs of the constructs, see *Table 2*. (**C**) Activities of different HUWE1 constructs at 5 µM concentration towards MCL1, monitored by anti-MCL1 Western blotting. For details, see Materials and methods.

The following figure supplement is available for figure 8:

**Figure supplement 1.** SAXS data for C-terminal HUWE1 constructs.

mixtures thereof. In line with our hypothesis, we find that the activation segment forms a complex with HUWE1 (3951–4374), as indicated by a marked shift of the elution peak toward larger volumes upon mixing the two components (*Figure 9B*). This interaction is specific to the activation segment, since it does not occur with MBP alone (*Figure 9C*). Furthermore, the interaction requires an intact dimerization region of HUWE1, since the removal of the pointer helix (HUWE1 [3975–4374]) or of the entire dimerization region (HUWE1 [3993–4374]) results in a loss of binding to the activation segment (*Figure 9D,E*).

In a control experiment addressing the rather small SEC-elution volume of the MBP-tagged activation segment (*Figure 9B,D,E*), we determined the MW of this construct by MALS and confirmed that it is monomeric (*Figure 9—figure supplement 1*). The relatively large hydrodynamic

**Table 2.** Summary of SEC-MALS-derived MWs and SAXS-derived $R_g$-values for C-terminal HUWE1 constructs. For SEC-MALS studies the proteins were injected at a concentration of ~375 µM. The SAXS experiments were performed at ~40 µM concentration (see *Figure 8—figure supplement 1*).

| HUWE1 construct | MW (MALS) / kDa | MW (calc.) / kDa | $R_g$ (SAXS) / Å |
|---|---|---|---|
| 3759–4374 | 68 | 74 | 35 |
| 3810–4374 | 61 | 68 | 35 |
| 3843–4374 | 60 | 65 | 31 |
| 3896–4374 | 92 | 60 | 39 |
| 3951–4374 | 93 | 53 | 36 |
| 3975–4374 | 51 | 50 | 27 |
| 3993–4374 | 44 | 47 | / |

radius of this construct, hence, originates from an extended shape of the activation segment (76 residues, including a His$_6$-tag and a protease cleavage site) that is fused to MBP.

Taken together, these experiments provide evidence for a specific interaction between the activation segment and the C-terminal region of HUWE1 that is detectable in trans and dependent on the dimerization region. The conformational state and the catalytic activity of HUWE1 may thus be controlled by a balance of inter- and intramolecular interactions.

## The HUWE1-inhibitor p14ARF interacts with the activation segment and promotes oligomerization of HUWE1

How inter- and intramolecular interactions compete in the structural regulation of HUWE1 will be determined by various factors, such as the local concentrations of HUWE1, the respective affinities of the interacting regions, and their structural accessibilities. All these factors may, in turn, be influenced by macromolecular interactions that stabilize particular conformational states of HUWE1. In this context, p14ARF is an interesting candidate, since it was shown to interact with HUWE1 directly and to inhibit its catalytic activity (through unknown mechanisms) (*Chen et al., 2005*).

p14ARF is a 14 kDa, highly basic protein with low overall sequence complexity and no conserved domains. In a series of binding and co-expression studies with truncated p14ARF variants, we identified a minimal region comprising residues 45–64 of p14ARF as a major binding site for HUWE1 (data not shown). To map the corresponding binding site for p14ARF on HUWE1, we monitored the interactions between a p14ARF-derived peptide (comprising residues 45–64) and our panel of HUWE1 constructs by fluorescence polarization (*Table 3*, *Figure 10A*). Interestingly, we find that all HUWE1 constructs that contain the activation segment, that is, HUWE1 (3759–4374), HUWE1 (3810–4374), and HUWE1 (3843–4374), bind to the p14ARF-peptide with dissociation constants in the low micromolar range (7.7, 6, and 5.2 µM, respectively). In contrast, shorter constructs of HUWE1 that are lacking the activation segment display significantly weaker affinities for p14ARF, with dissociation constants ranging from 230 to 340 µM. These results indicate that a major binding site for p14ARF resides either in the activation segment itself or in a conformational state of HUWE1, whose population depends on the activation segment. To discriminate between these possibilities, we determined the affinity of the p14ARF-peptide for the isolated activation segment (HUWE1 [3843–3902], tagged with MBP). The resulting dissociation constant, 3.6 µM, is similar to the ones determined for the activation segment in the context of the HECT domain. This implies that a major binding site for p14ARF on HUWE1 resides in the activation segment.

To illuminate the functional consequences of the interactions between p14ARF and HUWE1, we performed auto-ubiquitination assays with different HUWE1 constructs upon addition of a peptide comprising residues 45–75 of p14ARF (*Figure 10—figure supplement 1*). Analogous assays with the slightly shorter peptide, p14ARF (45-64), that was used in the fluorescence polarisation studies yielded very similar results (data not shown). Surprisingly, we find that the activities of HUWE1 (3843–4374), (3896–4374), and (3951–4374), are all significantly inhibited by the p14ARF-peptide. For the isolated HECT domain, no clear p14ARF-mediated inhibitory effect is observed. As noted above, the absolute ubiquitination levels vary between the different constructs, due to

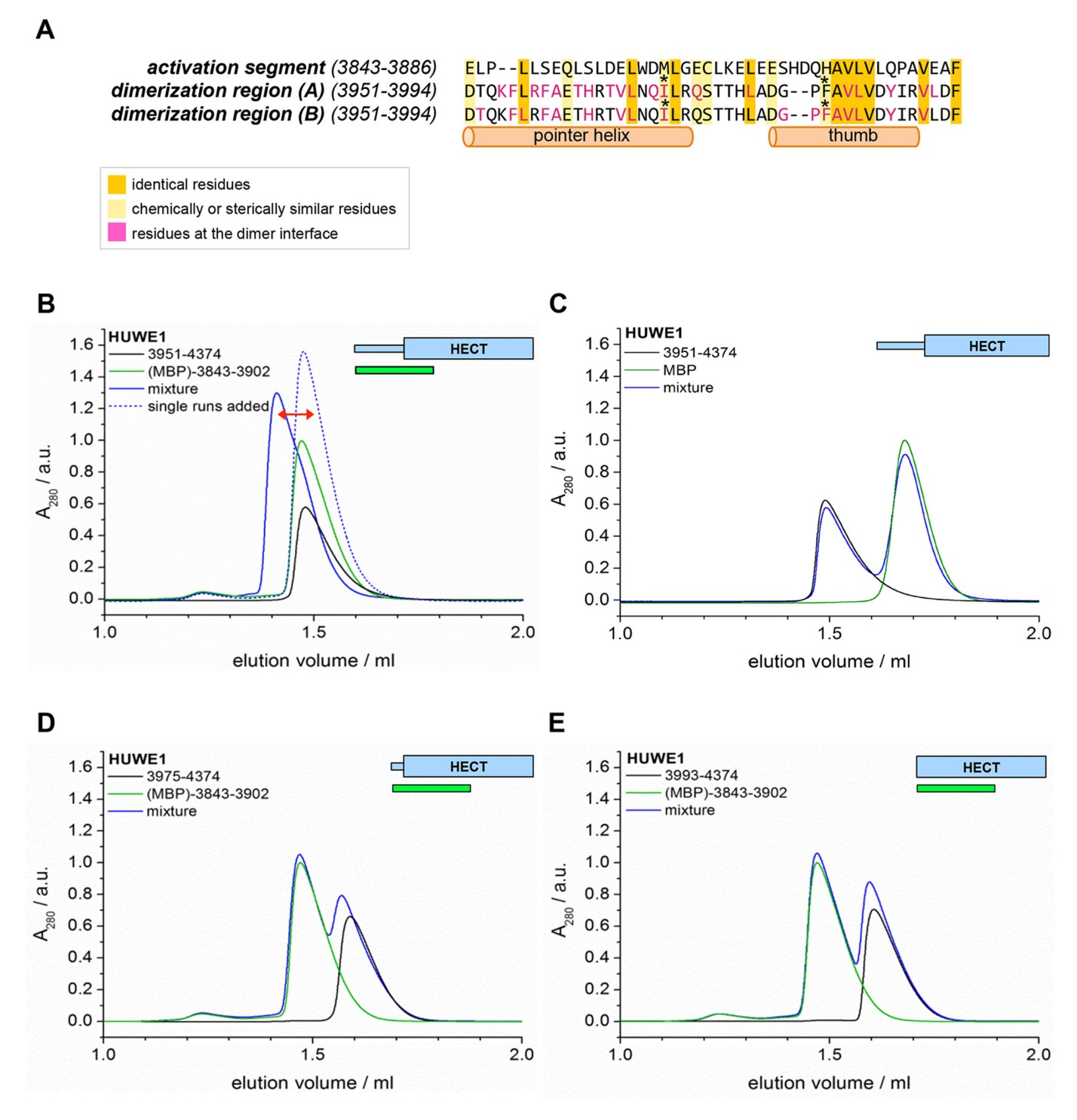

**Figure 9.** The activation segment displays sequence homology to the dimerization region and interacts with HUWE1 (3951–4374) specifically. (**A**) Sequence alignment of the activation segment with the dimerization region of HUWE1, generated by EMBOSS stretcher (*Rice et al., 2000*). Residues that form contacts at the dimer interface are shown in magenta. The mutations sites, Ile 3969 and Phe 3982, are marked by asterisks. (**B,D,E**) SEC experiments with different HUWE1 constructs, monitoring interactions with the MBP-tagged activation segment (residues 3843–3902) in trans. SEC profiles of the individual proteins at 120 μM concentration are shown in black and green, respectively; their 1.5:1 molar mixture is shown in blue. The peak heights for the activation segment alone (green) were normalized to a value of 1. In (**B**) a dashed line represents the expected elution peak for a mixture of non-interacting proteins. A red arrow marks the offset between the calculated and experimental peaks. No significant complex formation is detected in (**D**) and (**E**). (**C**) Analogous SEC experiment using MBP as a negative control.

*Figure 9 continued on next page*

*Figure 9 continued*

The following figure supplement is available for figure 9:

**Figure supplement 1.** The MBP-tagged activation segment of HUWE1 is monomeric.

variations in the numbers of ubiquitination sites and the distinct conformations of the constructs that likely impact the accessibilities of these sites.

Our studies thus show that the p14ARF-derived peptide - while interacting most strongly with the activation segment of HUWE1 - can, in principle, inhibit the auto-ubiquitination activity of HUWE1 in a manner that is independent of the activation segment. This, in turn, implies that the residual binding activity of p14ARF for the HUWE1 HECT domain is sufficient to confer inhibition, at least when studying truncated constructs of HUWE1 that are lacking the activation segment. In the context of longer HUWE1 constructs, however, it is the activation segment that confers tighter binding of p14ARF. This led us to speculate that p14ARF may impose a primary layer of regulation on HUWE1 by modulating its conformational equilibrium: By interacting with the activation segment, p14ARF may release this segment from its intramolecular engagement with the dimerization region and promote HUWE1 dimerization.

To test this idea, we subjected HUWE1 (3843–4374) that contains the activation segment to SEC experiments in the presence and absence of a four-fold molar excess of a p14ARF-derived peptide (residues 45–75) (*Figure 10B*). We find that the peptide co-elutes with HUWE1 (3843–4374), as shown by SDS-PAGE (*Figure 10B*, insert). Note that the peptide itself cannot be detected reliably by UV absorbance, due to a lack of aromatic amino acids ($\lambda = 280$ nm) and due to the analytical scale that these experiments are performed at ($\lambda = 220$ nm). Strikingly, the peptide triggers a marked shift in the SEC profile of HUWE1 (3843–4374), indicative of oligomerization. Shorter constructs of HUWE1 that lack the activation segment (HUWE1 [3896–4374], HUWE1 [3951–4374], and HUWE1 [3993–4374]) are not affected by the presence of the peptide, in line with the results of the fluorescence polarization studies (*Figure 10C–E*).

Taken together, these experiments are consistent with the intriguing notion that p14ARF may shift the conformational equilibrium of HUWE1 toward the inhibited dimeric state by capturing the activating segment. An additional/alternate mode of p14ARF-mediated inhibition of HUWE1 is independent of the activation segment. The structural underpinnings and relative contributions of these mechanisms in the regulation of HUWE1 remain to be elucidated.

## Discussion

In this study, we uncover a regulatory switch in the conformational equilibrium of the ubiquitin ligase HUWE1 and present—to our knowledge—the first crystal structure of a HECT E3 enzyme in an auto-

**Table 3.** Apparent dissociation constants, $K_D^{app}$, of the interactions between p14ARF and C-terminal HUWE1 constructs, as derived from fluorescence polarization studies with a fluorophor-labeled p14ARF-derived peptide (residues 45–64) (see *Figure 10A*).

| HUWE1 construct | $K_D^{app}/\mu M$ |
| --- | --- |
| 3759–4374 | 7.7 ± 0.4 |
| 3810–4374 | 6 ± 1 |
| 3843–4374 | 5.2 ± 0.2 |
| 3896–4374 | 340 ± 60 |
| 3951–4374 | 280 ± 40 |
| 3993–4374 | 230 ± 40 |
| (MBP-) 3843–3902 | 3.6 ± 0.2 |

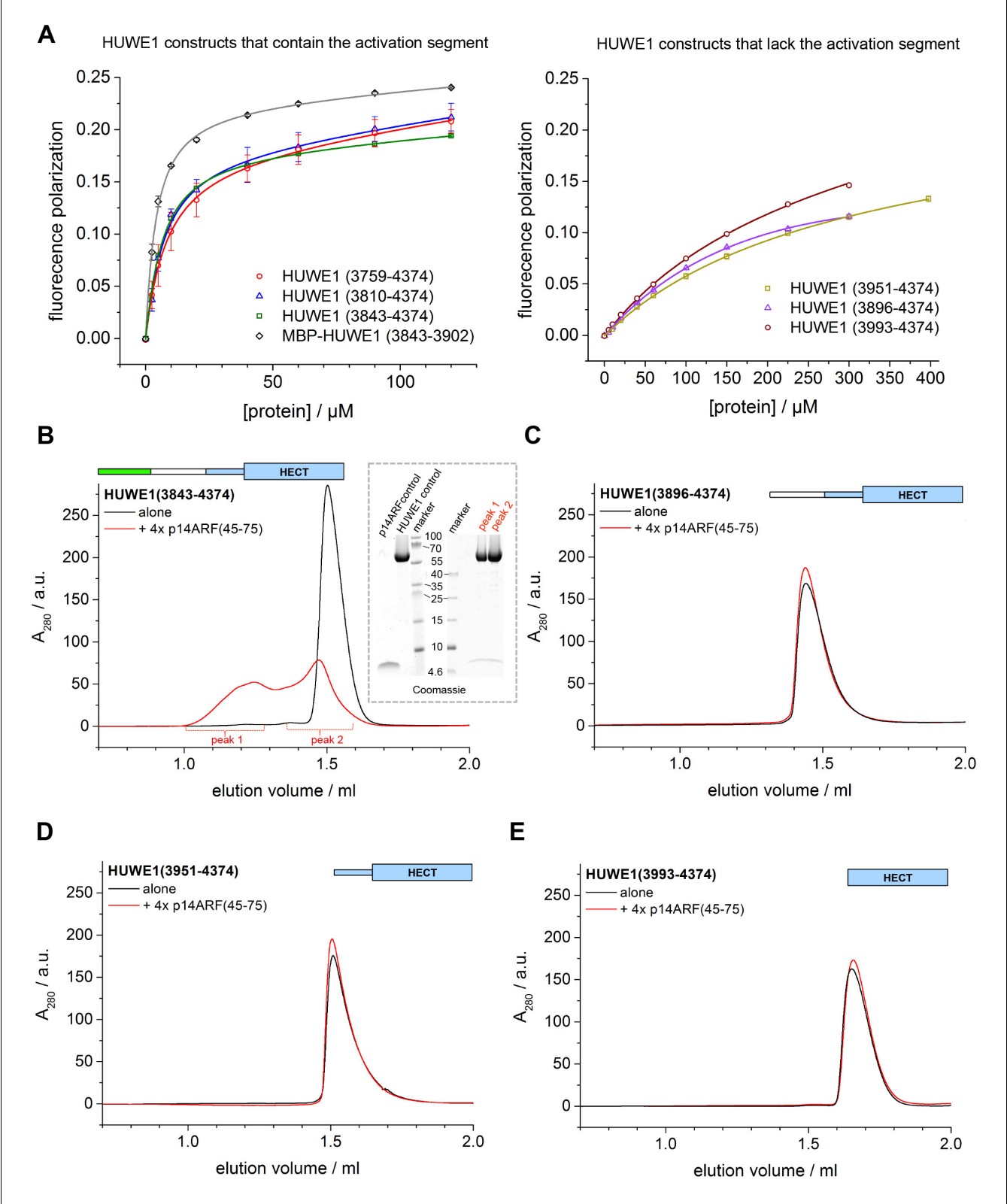

**Figure 10.** p14ARF interacts with the activation segment of HUWE1 and stimulates oligomerization of HUWE1. (**A**) Interactions of a fluorophor-labeled peptide comprising residues 45–64 of p14ARF with HUWE1 constructs of different lengths, monitored by fluorescence polarization. A constant peptide concentration of 1 µM was titrated with different concentrations of protein, as indicated. Constructs containing the activation segment, including the MBP-tagged activation segment per se are shown on the left; constructs lacking the activation segment on the right. Note that the x-axes have

*Figure 10 continued on next page*

*Figure 10 continued*

different scales. The data points and errors reflect the means and standard deviations obtained from three independent experiment replicates. Data were fitted (lines), as described in the Materials and methods section. The resulting $K_D^{app}$-values are summarized in *Table 3*. (B–E) SEC experiments with the specified constructs of HUWE1 at 120 µM concentration in the absence and presence of a four-fold molar excess of p14ARF (45-75). The p14ARF-derived peptide does not contain any aromatic residues and is, therefore, not detected by UV absorbance (λ = 280 nm). That HUWE1 (3843–4374) co-elutes with the p14ARF-derived peptide is demonstrated by SDS-PAGE, using 10–20% tricine gradient gels (insert). Two different molecular weight markers are shown. The HUWE1 bands look somewhat diffuse on this particular gel, since a prior pull-down using nonspecific protein-binding resin (Strataclean, Agilent) was performed in order to visualize the small amount of peptide.

The following figure supplement is available for figure 10:

**Figure supplement 1.** Effects of p14ARF (45-75) on the auto-ubiquitination activities of C-terminal HUWE1 constructs.

inhibited state. How the catalytic activities of HECT E3 enzymes are controlled is relatively poorly understood. A structural framework has so far only been delineated for members of the NEDD4-subfamily. These enzymes contain an N-terminal C2 domain that can serve an auto-inhibitory function by occupying a specific site on the N-lobe of the HECT domain that otherwise associates with ubiquitin and by structurally interfering with the transfer of ubiquitin from the E2 enzyme to the E3 active site cysteine (*Mari et al., 2014*; *Wiesner et al., 2007*; *Escobedo et al., 2014*; *Persaud et al., 2014*). This transfer step can also be antagonized by intramolecular interactions between the HECT domain and the preceding WW domains in NEDD4-type enzymes (*Riling et al., 2015*; *Gallagher et al., 2006*).

In contrast, we have discovered that the catalytic activity of HUWE1 is down-regulated by dimerization, as mediated by a previously uncharacterized region adjacent to the catalytic HECT domain. This surprising structural arrangement locks the C-lobe of one HUWE1 subunit conformationally in a position that is incompatible with catalysis for several reasons. Firstly, the C-lobe of HECT E3 enzymes requires flexibility with respect to the N-lobe for catalysis (*Verdecia et al., 2003*). In the context of the asymmetric HUWE1 dimer, however, the C-lobe of molecule B is tethered to the dimer interface and, therefore, immobilized. Secondly, the dimer interface buries a hydrophobic region on the C-lobe that was shown to interact with ubiquitin in E3 enzymes of the NEDD4-subfamily (*Maspero et al., 2013*; *Kamadurai et al., 2009*, *2013*). Thirdly, the C-tail, including a catalytically critical phenylalanine, −4 Phe (*Kamadurai et al., 2013*; *Salvat et al., 2004*), is occluded by the dimer interface.

We show that the dimerization of HUWE1 (3951–4374) is mediated by rather weak interactions. This suggests that the conformational ensemble of HUWE1 (3951–4374) is dynamic in solution and, hence, likely auto-inhibited overall through transient interactions between the C-lobes and the dimerization regions. In line with this idea, we observe a significant stimulation of activity upon disruption of the dimer interface, both in vitro and in cells, which indicates that the dimeric form of HUWE1 is auto-inhibited effectively. There is currently no indication that the HUWE1 dimer that has crystallized in an asymmetric form is also functionally asymmetric in solution, as was described for the U-box ligase CHIP (*Zhang et al., 2005*).

Our studies suggest that the dimerization of HUWE1 can be counteracted by an intramolecular engagement of the dimerization region with a segment of strikingly similar sequence, the 'activation segment', located some 50 residues upstream. We imagine that the activation segment serves as an intramolecular 'wedge' that disrupts the HUWE1 dimer when the ligase ought to be active. In line with the important functions of the dimerization region and the activation segment in controlling the conformational equilibrium of HUWE1, both regions are disproportionately highly conserved across the family of HUWE1 orthologues, including lower eukaryotic ones (*Figure 11*).

The identification of the activation segment and the resulting interplay of intra- and intermolecular interactions in the regulation of HUWE1 may explain an apparent inconsistency between our in vitro and cell-based studies (*Figures 6* and *7*): In both systems, the dimerization-deficient variants, I3969A and F3982A, display higher activities than the WT. Yet, the relative stimulatory effects of the mutations are different. When monitoring the auto-ubiquitination capacity of HUWE1 in vitro, the activity enhancement observed for the I3969A and F3982 variants compared to the WT is relatively similar. In the cell, however, the F3982A variant stimulates the ubiquitination of physiological

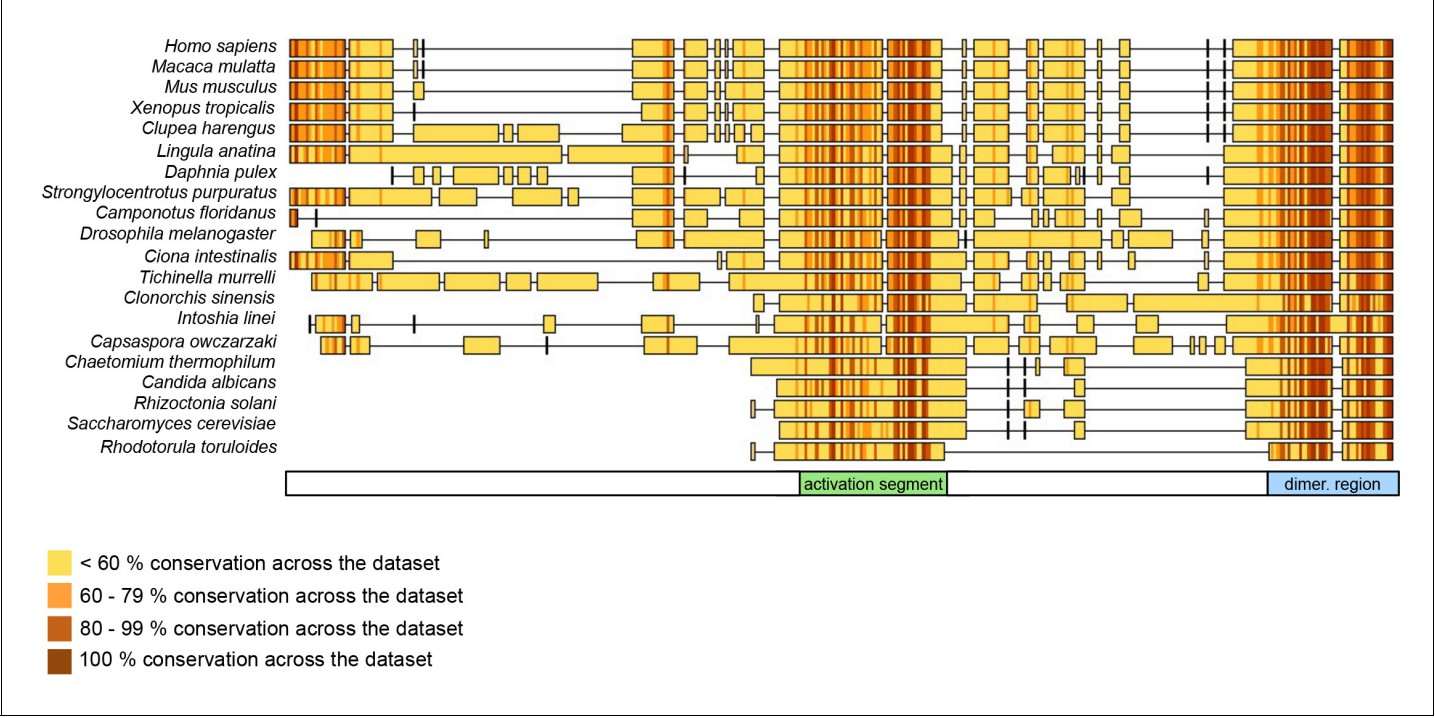

**Figure 11.** The dimerization region and the activation segment are disproportionally highly conserved across HUWE1 orthologues. Sequence alignment of 20 selected HUWE1 orthologues, as output by Clustal Omega (RRID:SCR_001591; http://www.ebi.ac.uk/Tools/msa/clustalo/) and sorted by their degree of sequence identity with the HECT domain of the human protein (top row). The illustration was prepared with Geneious Basic (RRID:SCR_010519) (*Kearse et al., 2012*). The alignment contains regions homologous to residues 3759 to 3992 of human HUWE1, that is, the region preceding the HECT domain.

HUWE1 substrates more strongly than the I3969A variant. Notably, our cell-based assays were performed with large HUWE1 constructs (2474–4374) that include the activation segment, while the in vitro assays were based on the crystallization construct, HUWE1 (3951–4374), that lacks the activation segment. It is possible that the mutated residues, both of which are pivotal for the integrity of the dimer interface, contribute differently to the alternate intramolecular interaction between the dimerization region and the activation segment. To characterize this latter interaction structurally and to delineate the conformational transition of HUWE1 from the dimeric, auto-inhibited to the monomeric, active state of HUWE1 will be an important aim of future studies.

It will also be essential to illuminate the physiological context, in which this transition occurs. With an in vitro dissociation constant of ~3 μM, the dimerization of the C-terminal region of HUWE1 (residues 3951–4374) may appear relatively weak. However, affinities in the low micromolar range (and even above) are not unusual in the ubiquitin system and have been observed, for instance, for the interactions between E2 and E3 enzymes (*Ye and Rape, 2009*; *Eletr and Kuhlman, 2007*), between ubiquitin and ubiquitin-binding domains (*Husnjak et al., 2012*), and between ubiquitin and E2 enzymes or E2/E3 complexes (*Buetow et al., 2015*; *Hofmann and Pickart, 2001*; *Wickliffe et al., 2011*). While these affinities, as determined in vitro, are typically altered by avidity and/or allovalency effects in the cellular context, it still emerges that ubiquitination enzymes have evolved to act through the sequential formation of rather weak protein interfaces. This feature may benefit the dynamic handover of reaction products throughout the catalytic cascade, the efficient assembly of ubiquitin chains, and the fine-tuned interplay of ubiquitination and deubiquitination enzymes.

Moreover, rather high overall cellular concentrations of HUWE1 have been reported, based on quantitative proteomics studies, for instance 120 nM in colon tissue (*Wiśniewski et al., 2015a*), 250 nM in colon cancer tissue (*Wiśniewski et al., 2015a*), and 400 nM in MCF7 cells (*Wiśniewski et al., 2015b*). Importantly, the local concentrations of HUWE1 at its cellular sites of action are expected to

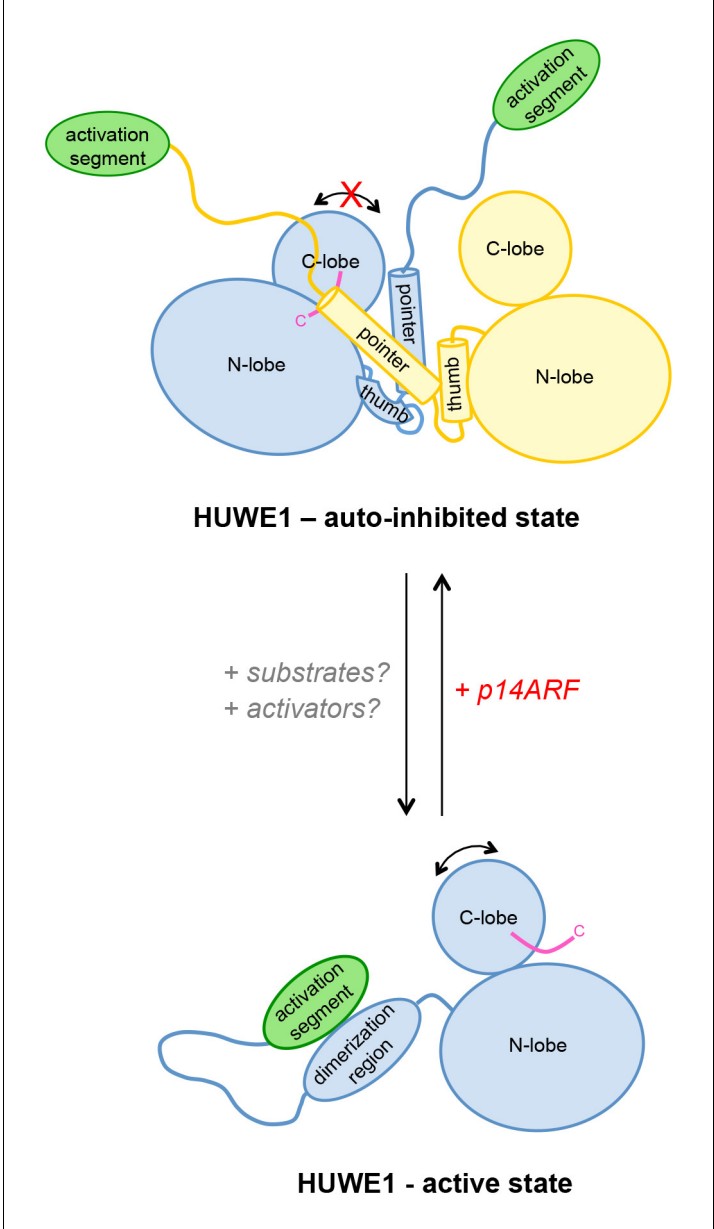

**Figure 12.** Model of the conformational regulation of HUWE1 and proposed mechanism of its inhibition by p14ARF. Our studies suggest that the catalytic activity of HUWE1 is regulated conformationally by an intricate balance of inter- and intramolecular interactions. The thumb and pointer helices adjacent to the catalytic HECT domain can mediate the dimerization of HUWE1. The dimer locks the position of the C-lobe, buries the C-terminal tail (magenta), and occludes a putative ubiquitin binding site on the C-lobe of one subunit, hence representing an auto-inhibited state. Alternatively, the dimerization region of HUWE1 can associate with the activation segment in cis, which precludes dimer formation. The activation segment and the dimerization region are separated by a 55-residue linker that is presumably flexible, thus allowing for the re-positioning of the activation segment. In the monomeric state of HUWE1, the C-lobe is mobile with respect to the N-lobe (arrow), and the C-terminal tail may anchor the C-lobe on the N-lobe or interact with substrates, as required for catalytic activity.

The activation segment of HUWE1 presents a major interaction site for a physiological inhibitor of HUWE1, p14ARF. We propose that the binding of p14ARF to the activation segment releases the dimerization region from its intramolecular engagement, thus shifting the conformational equilibrium of HUWE1 toward the auto-inhibited, dimeric state. An additional/alternate mode of inhibition of HUWE1 by p14ARF is independent of the activation segment. We speculate that interactions of HUWE1 with its substrates and positive effectors favor the active state of HUWE1 through interactions that remain to be elucidated.

be significantly higher, due to the effects of cellular compartimentalization and protein co-localization. Considering the large size of HUWE1 (482 kDa) and, accordingly, the large number of its interaction partners, co-localization effects likely have a particularly pronounced impact on the local concentrations of HUWE1 and, therefore, on its propensity to dimerize (*Ispolatov et al., 2005*). Hence, even seemingly weak interactions of HUWE1, as determined in vitro, can be expected to fall into a physiologically meaningful range, that is, a range that can give rise to a functionally relevant population of the dimeric state, yet that allows for this population to be regulated by cellular factors.

We propose that the conformational equilibrium and, hence, the activity of HUWE1 is modulated by yet uncharacterized domain interactions in the context of full-length HUWE1 and by macromolecular factors that will determine not only the local concentrations of HUWE1 in the cell, but also the accessibility of the dimerization region and the activation segment for their intramolecular engagement (*Figure 12*).

The HUWE1-inhibitor and tumor suppressor p14ARF presents such a macromolecular factor. Our studies reveal that p14ARF binds to the activation segment of HUWE1, which may release this segment from its intramolecular engagement with the dimerization region and, thus, promote dimerization. The fact that the interactions between p14ARF and HUWE1 and those mediating the dimerization of HUWE1 are in the same range $K_D$-range in vitro, makes it plausible that they can regulate each other dynamically. We also speculate that high levels of p14ARF may contribute to the robust self-association and auto-inhibition of HUWE1 that we observe in HeLa cells. Notably, however, our studies show that an alternate or additional mode of p14ARF-mediated inhibition of HUWE1 also exists and is independent of the activation segment. In order to understand these mechanisms in detail and to clarify their relative contributions to the inhibition of HUWE1 by p14ARF, structural insights into HUWE1/p14ARF complexes are required.

We predict that the activation segment provides interaction sites for other cellular factors, in addition to p14ARF. Interestingly, a binding site for PCNA ('PIP-box') has recently been mapped to residues 3880-3887 of HUWE1 (*Choe et al., 2016*), thus coinciding with the activation segment. Whether PCNA binding impacts the oligomeric state and activity of HUWE1 remains to be determined.

Taken together, our studies lay out an unprecedented mechanistic framework for the conformational regulation of HUWE1 and its interactions with the tumor suppressor p14ARF. These insights have important ramifications for our understanding of the pathophysiological mechanisms of altered HUWE1 variants and for the design of therapeutic avenues toward targeting HUWE1 function. For instance, several point mutations that were identified in human tumors fall into the dimerization region and result in non-conservative amino acid changes of interfacing residues. We show that two of these sites, His 3962 and Ile 3969, are critical for dimerization and for the auto-inhibition of HUWE1. These observations are in line with the notion that there may be selective pressure, at least in some human tumors, to enhance the activity of HUWE1 and to render it resistant to inhibition by p14ARF.

We anticipate that the dimerization of HUWE1 may be manipulated by small-molecules or peptide-based agents that influence the interactions of the dimerization region and the activation segment, respectively. Moreover, p14ARF-mimetic peptides may be used to inhibit HUWE1 activity in the cell. Anti-tumor characteristics have recently been reported for a p14ARF-derived peptide in gefitinib-resistant non-small cell lung cancers (*Saito et al., 2013*). While a connection to HUWE1 has not been established in this context, it is interesting to note that the critical p14ARF region identified in that study comprises residues 38–65 and, thus, includes the region of p14ARF (residues 45-64) that we found to interact with HUWE1 and inhibit its activity. Thus, with the dimerization region and the activation segment of HUWE1 and the HUWE1-binding region of p14ARF we have defined three molecular knobs, by which the conformational equilibrium and, hence, the activity of HUWE1 may be tuned for therapeutic purposes.

## Materials and methods

### Gene construct design

For the in vitro studies, the HUWE1 gene constructs (3759–4374, 3810–4374, 3843–4374, 3896–4374, 3951–4374, 3975–4374, and 3993–4374) were sub-cloned into a pBADM11 vector (EMBL, Heidelberg, Germany) encoding an N-terminal TEV protease-cleavable $His_6$-tag (sequence: MKHHHHHHPMSDYDIPTTENLYFQ). Mutations in the HUWE1 gene were introduced by site-directed mutagenesis and ligation-during-amplification approaches (*Chen and Ruffner, 1998*). The HUWE1 gene construct encoding residues 3843–3902 was sub-cloned into a pETM41 vector (EMBL, Heidelberg, Germany), encoding an N-terminal TEV protease-cleavable MBP-tag and a C-terminal 3C protease-cleavable $His_6$-tag. The constructs encoding human UBA1 and ubiquitin were described previously (*Wickliffe et al., 2011*). The human UBCH7 gene (kindly provided by Michael Rape, UC Berkeley, CA), was sub-cloned into a pSKB2 vector (pET-28a [Merck, Darmstadt, Germany], modified to encode an N-terminal 3C protease-cleavable $His_6$-SUMO-tag). A region encoding residues 26–325 of the human MCL1 gene (amplified from Addgene plasmid # 25375 (*Morel et al., 2009*); a gift from Roger Davis) was subcloned into a modified pSKB2 vector, in which the N-terminal $His_6$-tag was replaced by a lipoyl domain, comprising residues 2–85 of branched-chain alpha-keto acid dehydrogenase subunit E2 from *Geobacillus stearothermophilus* (*Hipps et al., 1994*), followed by a TEV protease-cleavage site. The construct further encodes C-terminal HA- and $His_6$-tags. All sub-cloning procedures were performed by ligation-free methods (*van den Ent and Löwe, 2006*).

For transient transfections the following constructs were utilized: pcDNA3-$His_6$-ubiquitin (*Peter et al., 2014*), pcDNA3-HA-HUWE1-ΔN (*Adhikary et al., 2005*), pcDNA3-HA-MCL1 (*Peter et al., 2014*), and pcDNA3-MYC (*Adhikary et al., 2005*). The pFLAG-HUWE1-ΔN vector was generated based on pcDNA3-HA-HUWE1-ΔN. To generate stable cell lines, HA-HUWE1-ΔN and FLAG-HUWE1-ΔN were sub-cloned into modified pRRL vectors (*Wiese et al., 2015*), using AgeI and SpeI restriction sites. Alternatively, the HA- and FLAG-tagged HUWE1 genes were sub-cloned into CMV-based transposon vectors carrying the puromycin resistance gene, kindly provided by Thorsten Stühmer (Comprehensive Cancer Center Mainfranken, Würzburg, Germany).

### Protein expression and purification

All C-terminal HUWE1 fragments were expressed in LOBSTR RIL cells (Kerafast, Boston, MA) at 15°C for 16 hr after induction with 0.05% L-arabinose. $His_6$-MBP-tagged HUWE1 (3843–3902) and lipoyl domain-/$His_6$-tagged MCL1 were expressed in *E.coli* BL21(DE3) and LOBSTR RIL cells, respectively, at 15°C for 16 hr after induction with 0.5 mM IPTG. Cells were lysed in 80 mM HEPES pH 8.0, 500 mM NaCl, 10% glycerol, 20 mM imidazole, 5 mM $\beta$-mercaptoethanol (lysis buffer), containing protease inhibitor cocktail (Sigma-Aldrich, St. Louis, MO). $His_6$- and $His_6$-MBP-fusion proteins were purified from the supernatant by IMAC (immobilized metal ion affinity chromatography) using a HisTrap HP column (GE Healthcare, Uppsala, Sweden; buffer A: lysis buffer; buffer B: lysis buffer +980 mM imidazole). For cleavage of the $His_6$-tag, the sample was buffer-exchanged at 4°C over night into 20 mM HEPES pH 8.0, 250 mM NaCl, 10 mM imidazole, and 3 mM $\beta$-mercaptoethanol in the presence of $His_6$-tagged TEV protease. The protease was subsequently removed by an additional IMAC step. Tagged and untagged proteins, respectively, were further purified by gel filtration (Superdex 200, GE Healthcare) in 20 mM HEPES pH 8.0, 150 mM NaCl, 1 mM EDTA, and 5 mM DTT.

UBCH7 was expressed in *E. coli* BL21(DE3) cells at 20°C in TB-medium for 20 hours after induction with 0.5 mM IPTG. Harvested cells were lysed in 500 mM NaCl, 50 mM Tris/HCl, pH 8.0, 4% glycerol, 0.3% Triton X-100, 8 mM $\beta$-mercaptoethanol, 5 mM benzamidine, containing protease-inhibitor cocktail (Sigma-Aldrich). The $His_6$-tagged protein was purified by IMAC using a HisTrap HP column (GE Healthcare; buffer A: 400 mM NaCl, 50 mM Tris/HCl, pH 8.0, 20 mM imidazole, 8 mM $\beta$-mercaptoethanol; buffer B: buffer A + 480 mM imidazole), dialyzed at 4°C in the presence of $His_6$-3C protease over night and subjected to a second IMAC step before gel filtration (Superdex 75, GE Healthcare) in 200 mM NaCl, 50 mM Tris/HCl, pH 7.5, and 2 mM DTT.

Human UBA1 and ubiquitin were prepared as described previously (*Wickliffe et al., 2011*).

## Synthetic peptides

Synthetic peptides corresponding to residues 45–64 and 45–75 of human p14ARF were purchased from Elim Biopharm (Hayward, CA) at >95% purity. For fluorescence polarization studies, a peptide comprising residues 45–64 of p14ARF was purchased with a TAMRA-label attached to the side chain of an additional lysine residue at the C-terminus of the peptide. In activity assays and SEC-studies, both p14ARF (45-64) and (45-75) behaved very similarly.

## Crystallization, data collection, and structure calculation

Crystals of HUWE1 (3951–4374) grew at 20°C in sitting drops containing 100 mM HEPES pH 7.0, 15% (w/v) PEG 20,000, and 100 mM glycine and were cryo-protected in the same solution, including 30% glycerol. Diffraction data were collected at the European Synchrotron Radiation Facility (ESRF), Grenoble, France, beamline ID30A-1 to 2.7 Å resolution and were processed with XDS (*Kabsch, 2010*). Molecular replacement was performed with Phaser (*McCoy et al., 2007*), as implemented in the collaborative computational project no. 4 (ccp4) suite (RRID:SCR_007255) (*Winn et al., 2011*), using a structure of the HUWE1 HECT domain as a search model (PDB ID: 3H1D) (*Pandya et al., 2010*). Refinement was performed using Phenix (RRID:SCR_014224) (*Adams et al., 2010*) with individual B-factors, TLS (translation/libration/screw) and torsion angle NCS (non-crystallographic symmetry) restraints. Manual model building was performed in Coot (RRID:SCR_014222) (*Emsley et al., 2004*). In molecule B, residues 4171, 4172, and 4191–4197 could not be modeled and the intervening region (4173–4190) has poor or missing side chain density. This region is, however, remote from the dimer interface that we interpret in this study. In molecule A, the C-tail (4370–4374) could not not be built.

## Small-angle X-ray scattering

SAXS data were collected at ESRF beamline BM29 using a Pilatus 1M detector (Dectris, Baden-Daettwil, Switzerland). Experiments were carried out at 10°C with HUWE1 constructs at various concentrations between 1 and 15 mg/ml in 20 mM HEPES pH 8.0, 150 mM NaCl, 1 mM EDTA, and 5 mM DTT. For each sample, 10 consecutive 1 s-exposures were compared, but no significant radiation damage was detected. Scattering data were reduced and processed using PRIMUS (*Konarev et al., 2003*), and the scattering profile of the buffer was subtracted. To estimate the radius of gyration ($R_g$) data from the samples at the lowest concentrations were subjected to AUTORG analysis (*Petoukhov et al., 2007*). This yielded very similar $R_g$ values to those derived from the pair distribution function, P(r), as calculated with the indirect transform package GNOM (*Semenyuk and Svergun, 1991*).

For the structure-based simulations of scattering profiles, we used the AllosMod-FoXS server (http://modbase.compbio.ucsf.edu/allosmod-foxs/) (*Schneidman-Duhovny et al., 2016*; *Weinkam et al., 2012*) in the static mode. Residues missing in the crystal structures were built automatically. The top fit scores for a single model and the corresponding $R_g$-values are quoted. To analyze simulations for a HUWE1 monomer with a flexible dimerization region, we made use of the MES (multiple ensemble search) (*Pelikan et al., 2009*) option of AllosModFoXS.

## Multi-angle light scattering

SEC-MALS was performed at room temperature using a Superdex 200 10/300 GL column (GE Healthcare) coupled to a Dawn 8+ MALS detector and Optilab T-rEX refractive index detector (Wyatt Technology, Santa Barbara, CA). Proteins were injected at the indicated concentrations in a buffer containing 20 mM HEPES pH 8.0, 250 mM NaCl, 1 mM EDTA, and 5 mM DTT. MWs were determined at the absorbance peak tips using the ASTRA 6 software (Wyatt Technology).

## Analytical size-exclusion chromatography

Proteins were injected onto a Superdex 200 Increase 3.2/300 column (GE Healthcare) at 4°C and at the indicated concentrations in 20 mM HEPES pH 8.0, 100 mM NaCl, 1 mM EDTA, and 5 mM DTT. When comparing HUWE1 in the presence and absence of the p14ARF-derived peptide, the samples contained 10% DMSO. Extensive control experiments over a range of DMSO concentrations were performed and ascertained that the DMSO content used here does not affect the oligomerization state of HUWE1.

Note that the Superdex 200 Increase 3.2/300 column has a significantly smaller bed volume (2.4 ml) than the Superdex 200 10/300 GL column (24 ml) that was used for SEC-MALS studies. This results in a smaller dilution of the protein samples during our SEC experiments compared to our SEC-MALS experiments. To account for the differences in dilution and injected volumes, we adjusted the protein concentrations accordingly: Protein concentrations of ~120 μM in SEC experiments yielded a comparable degree of dimerization of HUWE1 (3951-4374) as ~375 μM in SEC-MALS experiments.

## Analytical ultracentrifugation

Sedimentation equilibrium experiments experiments were conducted at 4°C using a ProteomeLab XL-A/XL-I ultracentrifuge (Beckmann Coulter, Brea, CA), equipped with an An-50 Ti rotor. 10 μM HUWE1 (3951–4374) in 10 mM HEPES pH 8.0, 200 mM NaCl, 1 mM EDTA, and 1 mM $\beta$-mercaptoe-thanol was filled in a 12 mm six-channel cell centrepiece and its distribution was followed by absorbance detection at $\lambda$ = 280 nm at rotation speeds of 9000 rpm, 13,000 rpm, and 17,000 rpm. The program Sedfit (*Schuck, 2000*) was used during the experiments to test whether an equilibrium was reached. The data were fitted using a monomer-dimer self-association model implemented in the program Sedphat (*Houtman et al., 2007*) with a monomer MW of 49.752 kDa.

## Fluorescence polarization

Fluorescence polarization measurements were performed at 25°C using a Clariostar microplate reader (BMG Labtech, Ortenberg, Germany), 540 nm excitation and 590 nm emission wavelengths, in 96-well flat-bottom microplates (Greiner Bio-One, Frickenhausen, Germany). Proteins and fluorophor-labeled p14ARF-derived peptide were in 20 mM HEPES pH 8.0, 150 mM NaCl, 10% glycerol, 1 mM EDTA, and 5 mM DTT. Titration points were prepared individually by mixing two stock solutions containing a constant peptide concentration of 1 μM and either none or the maximum concentration of protein. Average binding curves from three independent experiments were fitted using the formula

$$A = A_f + (A_b - A_f) \cdot \left( (K_D + L + x) - \frac{\sqrt{\left( (K_D + L + c)^2 - 4 \cdot c \cdot L \right)}}{2} \right) + t \cdot c$$

where $A$ is the anisotropy at protein concentration $c$, $A_b$ the anisotropy when the fluorophor is completely bound to protein, $A_f$ the anisotropy in the absence of protein, $K_D$ the dissociation constant, $L$ the concentration of labelled peptide, and $t$ a constant.

## Activity assays in vitro

To compare the auto-ubiquitination activities of HUWE1 (3951–4374) WT versus mutated variants, the constructs were incubated at the indicated concentrations with 200 nM E1 enzyme, 5 μM E2 enzyme (UBCH7), 500 μM ubiquitin, 3 mM ATP, and 8 mM MgCl$_2$. Substrate ubiquitination activities of HUWE1 (3951–4374), HUWE1 (3975–4374), and HUWE1 (3993–4374) at 5 μM concentration were compared under the same conditions, except for the presence of 60 μM MCL1. To monitor the effect of p14ARF on HUWE1 auto-ubiquitination activity, reaction mixtures contained p14ARF at 15 and 75 μM concentration, respectively, and 100 nM E1, 3 μM UBCH7, 5 μM HUWE1, 100 μM ubiquitin, 3 mM ATP, and 8 mM MgCl$_2$. Reactions were performed in 25 mM HEPES, pH 7.7 at 37°C for 15 min. They were started by the addition of ATP and quenched in reducing SDS sample buffer. Samples were subjected to 10% Tris-glycine SDS-PAGE. Western blots performed using the anti-ubiquitin antibody FK2 (Enzo Life Sciences, Exeter, UK; RRID:AB_10541840) or the anti-MCL1 antibody S-19 (Santa Cruz Biotechnology, Dallas, TX; RRID:AB_2144105). Experiments were performed as independent triplicates. The results were quantified with ImageJ (RRID:SCR_003070) (*Abràmoff and Magalhães, 2004*), when appropriate, and the mean and standard deviations were plotted.

## Cell-based assays

HeLa (RRID:CVCL_0030) and HEK293 (RRID:CVCL_0045) cell lines were obtained from ATCC (Manassas, VA), authenticated by STR profiling. Cells were cultured in DMEM medium supplemented with

10% FBS and 1% penicillin/streptomycin and were regularly tested for mycoplasma contamination, using the Venor GeM mycoplasma PCR detection kit (Minerva Biolabs, Berlin, Germany). Transient transfections with the indicated plasmids were conducted in six-well plates using polyethylenimine, according to standard protocols. Stable HeLa cell lines were generated by lentiviral transduction, as described previously (*Wiese et al., 2015*). For transposon-mediated gene expression HUWE1-expressing transposon vectors were co-transfected with the SB transposase-expressing vector (*Mátés et al., 2009*) into HeLa cells, using polyethylenimine, and stable cell pools were established by selection with puromycin.

For in vivo ubiquitination assays, cells were lysed by sonication in urea buffer (6 M urea, PBS, 10 mM imidazole), and lysates were cleared by centrifugation. Ubiquitinated proteins were captured on Ni-NTA resin at room temperature over night, washed in PBS buffer containing 35 mM imidazole and analyzed by immunoblotting with the following antibodies: anti-HA C29F4 (Cell Signaling Technology, Danvers, MA; RRID:AB_10693385), anti-C-MYC D3N8F (Cell Signaling Technology; RRID:AB_2631168), and anti-MCL1 S-19 (Santa Cruz Biotechnology; RRID:AB_2144105).

For immunoprecipitation, cells were lysed in TNT buffer (50 mM Tris pH 8.0, 250 mM NaCl, and 1% Triton X-100), supplemented with protease and phosphatase inhibitor cocktails (Sigma-Aldrich). Lysates were briefly sonicated and cleared by centrifugation. Protein complexes were recovered using protein G-agarose (Sigma-Aldrich) with anti-HA (C29F4, Cell Signaling Technology) or anti-FLAG (M2, Sigma-Aldrich) antibodies. Precipitated proteins were resolved on 10% SDS Bis-Tris gels and immunoblotted using the anti-HA (6E2, Cell Signaling Technology) and anti-FLAG (M2, Sigma-Aldrich) antibodies.

## Accession numbers

Atomic coordinates and structure factors have been deposited in the Protein Data Bank under accession code 5LP8.

## Acknowledgements

We thank Thorsten Stühmer (Comprehensive Cancer Center Mainfranken, University Hospital Würzburg) for sharing transposon expression vectors; the staff of ESRF beamlines ID30A and BM29; Julia Haubenreisser for technical assistance; Edward Lowe, Jeremy Thorner, Florian Sauer, Jacek Wisniewski, Hermann Schindelin, Stefanie Peter, Elmar Wolf, and Patrick Weinkam for helpful discussions; and Markus Seeliger for critical reading of the manuscript. While this paper was under editorial assessment, G. Prag and colleagues reported that Nedd4-subfamily ligases may also be regulated through oligomerization, however, by a mechanism that is different from HUWE1 (*Attali et al., 2017*).

## Additional information

### Funding

| Funder | Grant reference number | Author |
| --- | --- | --- |
| Deutsche Forschungsgemeinschaft | LO 2003/1-1 | Sonja G Lorenz |
| Wilhelm Sander-Stiftung | 2015.147.1 | Sonja G Lorenz |
| Deutsche Forschungsgemeinschaft | PO-1458/4-1 | Nikta Popov |
| Deutsche Forschungsgemeinschaft | Ei222/12-1 | Martin Eilers |

The funders had no role in study design, data collection and interpretation, or the decision to submit the work for publication.

### Author contributions

BS, Conceptualization, Investigation, Visualization, Writing—review and editing; WX, Investigation; ME, Writing—review and editing; NP, Supervision, Investigation, Writing—review and editing; SL,

Conceptualization, Supervision, Funding acquisition, Investigation, Visualization, Writing—original draft, Writing—review and editing

### Author ORCIDs

Sonja Lorenz, http://orcid.org/0000-0002-9639-2381

## Additional files

### Major datasets

The following dataset was generated:

| Author(s) | Year | Dataset title | Dataset URL | Database, license, and accessibility information |
|---|---|---|---|---|
| Sander B, Lorenz S G | 2016 | Crystal structure of an asymmetric dimer of the ubiquitin ligase HUWE1 | http://www.rcsb.org/pdb/explore/explore.do?structureId=5LP8 | Publicly available at the RCSB Protein Data Bank (accession no: 5LP8) |

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
