## [Decision Letter]

Thank you for submitting your article "A conformational switch regulates the ubiquitin ligase HUWE1" for consideration by *eLife*. Your article has been favorably evaluated by Michael Marletta as the Senior Editor and three reviewers, one of whom is a member of our Board of Reviewing Editors. The reviewers have opted to remain anonymous.

The reviewers have discussed the reviews with one another and the Reviewing Editor has drafted this decision to help you prepare a revised submission. We hope you will be able to submit the revised version within two months.

Summary:

The manuscript by Sander, et al., describes the partial of structure of the very large HUWE1 HECT E3 enzyme (nearly 500 kD), which includes the catalytic HECT domain and a region immediately upstream of the catalytic domain. While the structure of the HUWE1 catalytic domain, by itself, has been previously reported, the importance of the current structure is that it reveals a unique regulatory mechanism for this enzyme, mediated by the domains upstream of the HECT domain. The protein is shown to form an asymmetric dimer is a manner that is predicted to inhibit the activity of the enzyme, and this is confirmed biochemically. Most interesting, p14ARF, a known HUWE1-interacting protein, is proposed to stabilize the dimeric (inhibited) form of HUWE1.

The importance of this work is two-fold. First, the newly discovered mechanism of regulation of HUWE1 is so far unique among HECT E3s. This therefore adds to the list of surprisingly diverse mechanisms that regulate HECT E3s. Second, the HUWE1 and p14ARF are cancer-relevant proteins, and while it is not yet clear which substrate(s) of HUWE1 are most critical in cancers, the proposed mechanism of regulation by ARF will be an important contribution and lay the groundwork for further discovery.

While this manuscript reports interesting findings, several issues should be addressed in a revised version:

Essential revisions:

1) I am concerned about the physiological relevance of the dimerization model: The K_D_ in the μM range would require high concentrations of HUWE1 in the cells; what is known about HUWE1 levels and are they in the expected range for this model to be relevant? It would be essential that the authors determine the exact K_D_ rather than estimate it. In addition, in vivo evidence has only been provided using overexpressed HUWE1 – a problematic issue given the high K_D_ of dimer formation (i.e. it is likely that the authors force the enzyme into an inactive dimer-state by overexpressing it) – this would need to be repeated under better controlled conditions, for example in stable cell lines with both partners being expressed for short periods of time. I would assume that the in cis interaction between the activation segment and a dimerization helix would be strongly favored over the in trans interaction between two HECT-domains – this raises doubts about the relevance of the dimer interface.

2) In Figure 5, critical control experiments with the HECT domain (aa 3993-4374) and the I3969A/F3982A double mutant are missing as well as a concentration step where the DR-HECT construct can be safely assumed to be completely monomeric. Ideally, the construct with the truncated DR (aa 3975-4374, Figure 4, Figure 6) should also be included. As it is, Figure 5 is not conclusive since all constructs (also those that should be mostly monomeric) exhibit concentration dependent inhibition and none of them can be assumed to be completely active and monomeric. Moreover, the concentration effect on activity is far too large to be explained by a simple monomer-dimer equilibrium. If the monomeric form were 100% active and the dimeric 50% (as the authors suggest), the two-fold decrease in activity between 16 and 64 μM for the WT should coincide with a complete conversion from monomer to dimer. Given that the dimerization is weak and dynamic (see below), this is hardly possible within this concentration range.

In Figure 4 and Figure 5, the HUWE1 constructs contain the AS and thus should adopt an active conformation (see Figure 6 and Figure 9, Table 2). Therefore, mutations in the DR should not have an effect on activity. In line with this, in Figure 5 the WT is virtually as active as the I3969A mutant, but does not provide insight into the importance of dimerization for inhibition. On the other hand, differences in activities are observed in particular for the F3982A mutant contradicting that the AS activates the protein. In any case, these constructs cannot be compared to the in vitro assays due to the presence of the AS.

3) Along these lines, to determine whether mutations in the dimerization interface activate HUWE1, the authors should use a known substrate (in vitro), rather than autoubiquitylation, and they should perform the reaction at physiologically meaningful concentrations (not in the range of 16 μM!).

4) The DR-HECT construct crystallizes as an asymmetric dimer where one HECT domain adopts an active and the other an inhibited conformation through interactions with the DR. Importantly, the DR region scores very differently in the structure validation report. Secondly, the position of the C-lobe of molecule A "is determined by crystal contacts" and therefore may be precluded from binding the DR. These crystal contacts should be shown in a supplemental figure and the B-factors of the DR listed separately for molecules A and B in Table 1.

The fact that the dimer interaction is weak (see below) strongly suggests that the N-terminal extensions interact independently and transiently with the C-lobes in solution and not in a concerted manner that would inhibit one half of the dimer and leave the other half active as the authors suggest. The possibility that the HUWE1 DR-HECT may form a symmetric dimer in solution should be discussed upfront and throughout the paper unless the authors can rule out a symmetric dimer experimentally. If the DR-HECT construct is indeed asymmetric in solution the biological significance of a half-active enzyme should be discussed and compared to the CHIP E3 ligase where the asymmetry links a dimeric enzyme to monomeric Ub transfer (half-of-sites reactivity) (PMID: 16307917).

Subsection “The crystal structure of a C-terminal region of HUWE1 reveals an asymmetric dimer”, last paragraph: "[…] would thus, in principle, be free to swing […]". The authors should illustrate this by overlaying the structure of molecule A with the other structures of the HUWE1 HECT domain. The speculation that the C-lobe is free to move should be supported by data. Otherwise the mobility of the C-lobe is possible, but not founded on data.

The authors speculate that F4371 (-4F) being buried through interactions with the N-terminal end of the DR may form the basis for autoinhibition. Yet, the exact same DR region also contacts C-lobe residues (L4335, M4359, L4362) involved in thioester formation (PDB ID: 4bbn and 4lcd). The authors should clarify this experimentally.

5) There are some gaps in the p14ARF experiments that may require additional experiments. It is stated as "data not shown" that a region of p14ARF was identified that bound to HUWE1 (residues 43-75). It is further shown that a peptide corresponding to this region stabilized the dimeric form of a dimerization-competent HUWE1 construct, but had no effect on the isolated HUWE1. This implies that p14ARF, as well as the p14ARF peptide, binds to HUWE1 outside of the HECT domain, but this does not appear to have ever been tested, either here or in the literature. Given the reagents that the authors have in-hand, it seems that it would be very straight-forward to determine the binding site on HUWE1 for p14ARF. Finally, as stated in the Discussion, tumor-associated mutations have been identified in HUWE1 that fall within the dimerization domain. The prediction is that these mutations might result in hyper-activation of HUWE1 by inhibiting formation of the dimer. This result would add significantly to the impact of the current paper and it seems that this could be tested with relatively little additional effort.

---

## [Author Response]

*Essential revisions:*

*1) I am concerned about the physiological relevance of the dimerization model: The* K_D_*in the μM range would require high concentrations of HUWE1 in the cells; what is known about HUWE1 levels and are they in the expected range for this model to be relevant? It would be essential that the authors determine the exact* K_D_*rather than estimate it. In addition, in vivo evidence has only been provided using overexpressed HUWE1 – a problematic issue given the high* K_D_*of dimer formation (i.e. it is likely that the authors force the enzyme into an inactive dimer-state by overexpressing it) – this would need to be repeated under better controlled conditions, for example in stable cell lines with both partners being expressed for short periods of time.*

In response to this comment, we have now determined the dissociation constant, K_D_, for the dimerization of the C-terminal region of HUWE1 (residues 3951-4374) by analytical ultracentrifugation (AUC) (Figure 3; subsection “The C-terminal region of HUWE1 dimerizes in solution”, last paragraph) and obtained a value of 3 µM. That the dimerization of HUWE1 is physiologically relevant is supported by the following observations:

i) Quantitative proteomics studies performed in Matthias Mann’s laboratory have yielded relatively high overall concentrations of HUWE1 in human tissues, for instance 120 nM in colon (Wisniewski J R et al., J. Proteome Res. 14(9), 2015). Interestingly, HUWE1 levels are even higher in colon cancer tissue, where they amount to 250 nM (Wisniewski J R et al., J. Proteome Res. 14(9), 2015). Elevated HUWE1 levels have also been detected in human cancer cell lines, such as K562 (170 nM) and MCF7 (400 nM) (Wisniewski J R et al., Proteomics 15(7), 2015). Importantly, the *local* concentrations of HUWE1 at its cellular sites of action are expected to be significantly higher, due to the effects of cellular compartmentalization and protein co-localization (for a rationalization of this concept, see Pawson T & Scott J S, Science 78(5346), 1997 or Kuriyan J & Eisenberg D, Nature 450(7172), 2007). Considering the large size of HUWE1 (482 kDa) and, accordingly, the large number of its interaction partners, co-localization effects likely have a particularly pronounced impact on the local concentrations of HUWE1 and, therefore, on its propensity to dimerize (Ispolatov I et al., Nuc. Acids Res. 33(11), 2005).

Based on these data and considerations (that we have included in the Discussion), we expect even a low μM K_D_-value for HUWE1 dimerization, as measured in vitro, to fall within a physiologically relevant range, i.e. a range that can give rise to a functionally significant population of the dimeric state, yet that allows for this population to be regulated by cellular factors.

ii) Importantly, we have performed additional co-IP experiments from two cell lines stably expressing HUWE1, as the reviewers suggested (Figure 5; subsection “Key contacts in the crystallographic dimer interface mediate the dimerization of HUWE1 in vitro and its self-association in cells”, eighth paragraph). We show that the HUWE1 levels in one of these cells lines are significantly lower than upon transient expression (Figure 5—figure supplement 1). Nevertheless, HUWE1 self-associates at these concentrations robustly (Figure 5). Hence, our studies provide strong evidence that HUWE1 can dimerize in the cell.

iii) The low μM K_D_-value that we have determined describes the dimerization capacity of an isolated ~ 50 kDa-fragment of HUWE1 in vitro. It is likely that additional domain interactions of HUWE1, its association with macromolecular interaction partners, or posttranslational modifications contribute to the dimerization of full-length HUWE1 in the cell, thus lowering the effective K_D_. With the tumor suppressor p14ARF, intriguingly, we have identified one such macromolecular factor. We show that p14ARF interacts with the activation segment of HUWE1 (Figure 10; Table 3; see also Point 5) and propose that this interaction shifts the conformational equilibrium of HUWE1 towards the inhibited dimer (Figure 10). Interestingly, the K_D_-values for the dimerization of HUWE1 and for its interaction with p14ARF are in the same (low μM) range in vitro(Figure 3; Figure 10; Table 3). This makes it plausible that these two interactions may be able to regulate each other dynamically in the cell. These points have been added to the text (subsection “The HUWE1-inhibitor p14ARF interacts with the activation segment and promotes oligomerization of HUWE1”, second paragraph;– Discussion, ninth paragraph).

iv) It is worth noting that dissociation constants in the μM range (and even weaker), as measured in vitro, are common in the ubiquitin system. This holds, for instance, for the interactions between E2 and E3 enzymes (Ye Y & Rape M, Nat. Rev. Mol. Cell Biol. 10(11), 2009), between ubiquitin and ubiquitin-binding domains (Husnjak K & Dikic I, Annu. Rev. Biochem. 81, 2012), and for the interactions between ubiquitin and E2s or E2/E3 complexes (Buetow L et al., Mol. Cell 58(2), 2015; Hofmann R M & Pickart C M, J. Biol. Chem. 276(30), 2001; Wickliffe K et al., Cell 144(5), 2011). While the affinities determined in vitroare typically altered by avidity and/or allovalency effects in the cellular context, it still emerges that ubiquitination enzymes have evolved to act through the sequential formation of rather weak protein interfaces. This feature may benefit the dynamic handover of reaction products throughout the catalytic cascade, the efficient assembly of ubiquitin chains, and the fine-tuned interplay of ubiquitinating and deubiquitinating reactions. We have included a corresponding comment into the text (Discussion, sixth paragraph).

Taken together, we provide strong evidence that the dimerization of HUWE1 that we observe in vitroand that is mirrored by our cell-based experiments represents a physiologically relevant regulatory mechanism, i.e. a mechanism that occurs in an affinity window that is suitable for a modulation by physiological factors.

*I would assume that the in cis interaction between the activation segment and a dimerization helix would be strongly favored over the in trans interaction between two HECT-domains – this raises doubts about the relevance of the dimer interface.*

How *cis* and *trans* interactions of the dimerization region compete in the dynamic regulation of HUWE1 is a fascinating question and will depend, for one part, on the local concentrations of the respective binding elements (dimerization region and activation segment). Based on this, one would typically expect an intramolecular interaction of the dimerization region with the activation segment to prevail upon dimer formation. Remarkably, however, our cell-based studies demonstrate that long HUWE1 constructs that contain the activation segment self-associate and that this process is mediated by the same key contacts that mediate dimerization of short HUWE1 constructs in vitroand that we had predicted, based on the crystal structure (Figure 5).

It is important to note that the conformational equilibrium of HUWE1 not only depends on the activation segment and the dimerization region, but also on their accessibilities for interacting with each other. If the activation segment is engaged elsewhere, dimer formation can be favored. Indeed, our data provide evidence for such a regulatory mechanism: As suggested by the reviewers (Point 5), we mapped the binding site of p14ARF on HUWE1 by fluorescence polarization studies and reveal that the primary binding site for p14ARF on HUWE1 resides in the activation segment (Figure 10; Table 3).

We, therefore, propose that p14ARF may shield the activation segment from interacting with the dimerization region intramolecularly, thereby promoting dimer formation. We speculate that other macromolecular factors, besides p14ARF, impact the conformational equilibrium of HUWE1 by favoring or disfavoring dimer formation (Figure 12). This could occur through interactions with the dimerization region, with the activating segment, or with surfaces that are specific to the dimer or monomer. Furthermore, yet uncharacterized domain interactions within HUWE1 may impact its dimerization propensity. To identify such mechanisms and to elucidate structures of full-length HUWE1 will be an important area of future studies. We have included the above data and ideas into the text (subsection “The HUWE1-inhibitor p14ARF interacts with the activation segment and promotes oligomerization of HUWE1”, second paragraph; Discussion, ninth paragraph).

*2) In Figure 5, critical control experiments with the HECT domain (aa 3993-4374) and the I3969A/F3982A double mutant are missing as well as a concentration step where the DR-HECT construct can be safely assumed to be completely monomeric. Ideally, the construct with the truncated DR (aa 3975-4374, Figure 4, Figure 6) should also be included. As it is, Figure 5 is not conclusive since all constructs (also those that should be mostly monomeric) exhibit concentration dependent inhibition and none of them can be assumed to be completely active and monomeric. Moreover, the concentration effect on activity is far too large to be explained by a simple monomer-dimer equilibrium. If the monomeric form were 100% active and the dimeric 50% (as the authors suggest), the two-fold decrease in activity between 16 and 64 μM for the WT should coincide with a complete conversion from monomer to dimer. Given that the dimerization is weak and dynamic (see below), this is hardly possible within this concentration range.*

In response to this comment, we have performed additional activity assays, including the HECT domain of HUWE1, the I3969A/F3982A variant, and the truncated variant (3975-4374) (Figure 6, formerly Figure 5). We have also included the tumor-associated variant H3962D of HUWE1 (3951-4374), as suggested in Point 5, and have shifted the protein concentrations to a significantly lower regime of 1, 5, and 10 µM. These values were chosen in consideration of the K_D_ of dimerization, 3 µM, as determined by AUC (Figure 3; see Point 1), and of the physiological levels of HUWE1 (120-400 nM *overall* cellular concentration, as reported by quantitative proteomics, see Point 1).

The comparison of the activities of HUWE1 (3951-4374) WT to the panel of dimerization- deficient variants reveals that the mutated variants are more active than the WT (Figure 6). The stimulation is most pronounced for the double mutant, in line with the loss of dimerization that we observed for this variant (Figure 4). Notably, at these concentrations, only the activity of the WT displays a weak concentration dependence (Figure 6). Qualitatively, these effects confirm our initial findings and support our model that the dimerization of HUWE1 is auto-inhibitory. We have included a corresponding description in the text (subsection “The disruption of the dimerization interface enhances the activity of HUWE1 in vitro and in cells”, first paragraph). As explained below (Point 4), we have also clarified the text to state that, due to the transient and dynamic nature of dimerization in solution, the ensemble of HUWE1 molecules is most likely auto-inhibited overall (see the aforementioned paragraph).

“[…] *Ideally, the construct with the truncated DR (aa 3975-4374, Figure 4, Figure 6) should also be included.”*

When comparing the dimerization-dependent activities of HUWE1 constructs of different lengths quantitatively, auto-ubiquitination assays are hard to interpret, since the constructs have different numbers of ubiquitination sites and adopt distinct conformations and, therefore, display different auto-ubiquitination activities per se(subsection “The disruption of the dimerization interface enhances the activity of HUWE1 in vitro and in cells”, second paragraph). In response to this comment, we, therefore, set up a substrate ubiquitination assay and compared the activities of HUWE1 (3951-4374), HUWE1 (3975-4374) and (3993-4374) towards MCL1 (Figure 6). This assay shows that the truncated, dimerization-deficient constructs (HUWE1 (3975-4374) and (3993-4374)) are significantly more active than HUWE1 (3951-437) that contains an intact dimerization region, in line with our model. A description has been added to the text (see the aforementioned paragraph).

*In Figure 4 and 5BC, the HUWE1 constructs contain the AS and thus should adopt an active conformation (see Figure 6 and Figure 9, Table 2). Therefore, mutations in the DR should not have an effect on activity. In line with this, in Figure 5 the WT is virtually as active as the I3969A mutant, but does not provide insight into the importance of dimerization for inhibition. On the other hand, differences in activities are observed in particular for the F3982A mutant contradicting that the AS activates the protein. In any case, these constructs cannot be compared to the in vitro assays due to the presence of the AS.*

We thank the reviewers for this comment, which we have addressed in the text as follows: Indeed, the extended HUWE1 constructs employed in our cell-based studies contain the activation segment. Yet, we find that they are auto-inhibited (Figure 7, formerly Figure 5) in HeLa cells and that the WT constructs self-associate (Figure 5, formerly Figure 4). These observations suggest that additional factors regulate HUWE1 activity and stabilize the HUWE1 dimer in the cellular context. As the presence of the activation segment *per se* does not render HUWE1 fully active in HeLa cells, we speculate that the activation segment serves as a recognition site for cellular factors that impact the conformational equilibrium and, hence, the activity of HUWE1 (Discussion, eighth paragraph).

Interestingly, our studies identify one such factor, the tumor suppressor p14ARF. We have now included fluorescence polarization experiments (see Point 5) and show that the major binding site for p14ARF resides in the activation segment of HUWE1 (Table 3; Figure 10). We also provide initial evidence that the interaction between p14ARF and the activation segment impacts the conformational equilibrium of HUWE1 and promotes oligomerization in vitro(Figure 10). We speculate that high levels of p14ARF may contribute to the relatively low activity of HUWE1 that we observe in HeLa cells (Discussion, eighth paragraph). As noted above (Point 1), other intra-and intermolecular interactions of HUWE1 may also have a role in promoting its dimerization in the cell.

“In line with this, in Figure 5 the WT is virtually as active as the I3969A mutant, but does not provide insight into the importance of dimerization for inhibition. On the other hand, differences in activities are observed in particular for the F3982A mutant contradicting that the AS activates the protein.”

We have acknowledged the fact that the F3982A mutation has a stronger stimulating effect on substrate ubiquitination than the I3969A mutation in our cell-based assays, and we provide a possible rationale for this phenomenon (Discussion, fifth paragraph). In brief, we speculate that two residues, Phe 3982 and Ile 3969, both of which are important for HUWE1 dimerization (Figure 4, Figure 5) may differ in their contributions to the interaction between the dimerization region and the activation segment. For instance, it is conceivable that the substitution of Ile 3969, while destabilizing the dimer interface, also destabilizes the alternate interaction of the dimerization region with the activation segment. As a result, this substitution may impact activity only moderately in the context of long HUWE1 constructs. To test such possible mechanisms and to characterize the interaction between the dimerization region and the activation segment structurally is an important aim of our future studies.

*3) Along these lines, to determine whether mutations in the dimerization interface activate HUWE1, the authors should use a known substrate (*in vitro*), rather than autoubiquitylation, and they should perform the reaction at physiologically meaningful concentrations (not in the range of 16 μM!).*

In response to this comment, we have now included substrate ubiquitination assays in our study, monitoring the modification of MCL1 (Figure 6, Figure 8). The results of these assays demonstrate that HUWE1 constructs that dimerize (HUWE1 (3951-4374) and HUWE1 (3896- 4374)) are significantly less active than those constructs that are deficient in dimerization (either due to the lack of the dimerization region (Figure 6) or due to the presence of the activation segment (Figure 8)) These results support our model that the dimerization of HUWE1 presents an auto-inhibitory mechanism. Descriptions have been included in the text (subsection “The disruption of the dimerization interface enhances the activity of HUWE1 in vitro and in cells”, second paragraph; subsection “The dimerization capacity of HUWE1 is modulated by intramolecular interactions”, fourth paragraph).

In the context of this reconstituted biochemical assay, it is hard to assess what the physiologically most meaningful concentrations of HUWE1 are, since only small portions of the full-length protein are studied, in highly purified form, in a defined posttranslational modification state, in homogeneous distribution, and in the absence of interaction partners (other than E1, E2, ubiquitin, and substrate). In light of these considerations, we have chosen a HUWE1 concentration of 5 μM, i.e. slightly above the K_D_ of dimerization (3 μM) that we have determined in vitro(Figure 3). Encouragingly, the results of our in vitroassays are corroborated by our cell-based analyses (Figure 5; Figure 7).

*4) The DR-HECT construct crystallizes as an asymmetric dimer where one HECT domain adopts an active and the other an inhibited conformation through interactions with the DR. Importantly, the DR region scores very differently in the structure validation report. Secondly, the position of the C-lobe of molecule A "is determined by crystal contacts" and therefore may be precluded from binding the DR. These crystal contacts should be shown in a supplemental figure and the B-factors of the DR listed separately for molecules A and B in Table 1.*

In response to this comment, we now provide detailed illustrations and analyses of the lattice environment of the C-lobe of molecule A (Figure 2—figure supplement 3; subsection “The crystal structure of a C-terminal region of HUWE1 reveals an asymmetric dimer”, last paragraph). All of the lattice interfaces that the C-lobe is involved in are small and polar compared to the dimer interface that we have characterized and validated in solution. Importantly, we observe a HUWE1 dimer, but no higher–order oligomers in solution, even at high protein concentrations, and we show that the dimerization in solution can be disrupted by mutations in the crystallographic dimer interface (Figure 3–Figure 5). Taken together, these observations demonstrate that the polar lattice contacts shown in Figure 2—figure supplement 3 are not relevant in solution.

We thank the reviewers for pointing out the difference in the B-factors and address this comment by providing a comprehensive analysis, including a table with B-factors listed per chain, and for the HECT domains and dimerization regions separately (Figure 1—figure supplement A).

The B-factors for molecule B are generally higher than for chain A, but within an acceptable range of the Wilson B-factor. We rationalize this phenomenon by the fact that molecule B has a significantly different packing environment form molecule A: A significant portion of the HECT domain of chain B faces a large water channel, and this portion displays particularly high B-factors (Figure 1—figure supplement B). In contrast, molecule A is embedded into lattice contacts more tightly.

The dimerization region of molecule B has lower B-factors, on average, than the HECT domain, consistent with it being distant from the water channel. In molecule A, the dimerization region has higher B-factors than the corresponding HECT domain. Importantly, however, the dimerization regions in both molecules are well defined in the electron density and could be modeled with confidence.

*The fact that the dimer interaction is weak (see below) strongly suggests that the N-terminal extensions interact independently and transiently with the C-lobes in solution and not in a concerted manner that would inhibit one half of the dimer and leave the other half active as the authors suggest. The possibility that the HUWE1 DR-HECT may form a symmetric dimer in solution should be discussed upfront and throughout the paper unless the authors can rule out a symmetric dimer experimentally. If the DR-HECT construct is indeed asymmetric in solution the biological significance of a half-active enzyme should be discussed and compared to the CHIP E3 ligase where the asymmetry links a dimeric enzyme to monomeric Ub transfer (half-of-sites reactivity) (PMID: 16307917).*

We thank the reviewers for this insightful comment that we fully agree with and that we have now incorporated in the text explicitly (The disruption of the dimerization interface enhances the activity of HUWE1 in vitro and in cells”, first paragraph; Discussion, third paragraph). We have clarified the text to state that, while the nature of the interface observed crystallographically is asymmetric, we predict that the ensemble of molecules in solution is overall auto-inhibited, due to the transient and dynamic nature of the interface. This interpretation is supported by the results of our activity assays that demonstrate that the dimerization of HUWE1 is associated with auto-inhibition.

We have also added a comment to acknowledge the intriguing work on the structurally and functionally asymmetric U-box ubiquitin ligase by Laurence Pearl and colleagues and to contrast it with the case of HUWE1 (Discussion, third paragraph).

*The authors speculate that F4371 (-4F) being buried through interactions with the N-terminal end of the DR may form the basis for autoinhibition. Yet, the exact same DR region also contacts C-lobe residues (L4335, M4359, L4362) involved in thioester formation (PDB ID: 4bbn and 4lcd). The authors should clarify this experimentally.*

We thank the reviewers for pointing out the interesting possibility that the engagement of residues L4335, M4359, L4362 at the dimer interface may contribute to the auto-inhibited nature of this state. The hydrophobic character of two of these residues (L4335 and L4362) is conserved in NEDD4-type E3s (corresponding to L861 and M888 of NEDD4 (PDB ID: 4BBN; Maspero E et al., Nat. Struct. Mol. Biol. 20(6), 2013), L916 and M943 of NEDD4L (PDB ID: 3JVZ; Kamadurai H B et al., Mol. Cell 36(6), 2009), and L771 and L798 of RSP5 (PDB ID: 4LCD; Kamadurai H B et al., e*Life* 2:e00828, 2013)). The third residue (M4359 of HUWE1) is substituted by a lysine residue in NEDD4-type E3s. The two conserved hydrophobic residues were shown to have a role in the transfer of ubiquitin from E2 to NEDD4-family E3s (Kamadurai H B et al., Mol. Cell 36(6), 2009; Kamadurai H B et al. e*Life* 2:e00828, 2013), since they provide contacts with donor ubiquitin.

It is currently unknown if HUWE1 positions the donor ubiquitin on the C-lobe in a similar manner. Unfortunately, it is problematic to test the above hypothesis in functional assays for the following reason: As noted by Hidde Ploegh and colleagues for HUWE1 (Pandya R K et al., J. Biol. Chem. 285(8), 2010), and in line with studies on other HECT E3 enzymes (e.g. Mari S. et al., Structure 22(11), 2014; Maspero E. et al., EMBO Rep. 12(4), 2011), the thioesterification of ubiquitin on the active site of HECT E3s is difficult to visualize, due to rapid isopeptide bond formation. To circumvent this problem, previous studies, including the one on HUWE1, have typically used truncated constructs of the HECT domain, in which the C-tail, including the conserved ‘-4 Phe’, was deleted or mutated. Such HECT constructs show impaired isopeptide bond formation with ubiquitin, while thioester formation is uncompromised, and thus allow for thioester formation to be monitored (Salvat C et al., J. Biol. Chem. 279(18), 2004). However, when studying the inhibitory mode of the HUWE1 dimer, we cannot make use of such an approach, since the C-tail of HUWE1 makes up an integral part of the dimer interface on molecule B in our structure (Figure 2). So, unfortunately, we are unable to provide an experimental dissection of the inhibitory contributions of the various possible mechanisms at this point in time. However, we will try to study this interesting question in the future.

We incorporated the reviewers’ idea into the text (subsection “The crystal structure of a C-terminal region of HUWE1 reveals an asymmetric dimer”, fifth paragraph; Discussion, third paragraph) as a third rationale, for why the HUWE1 dimer is inhibited, in addition to the two mechanisms that we had suggested (lack of C-lobe mobility and burial of the C-tail). It is possible that all three mechanisms apply to inhibit HUWE1 synergistically. We now also provide an additional figure (Figure 2—figure supplement 1) to illustrate that donor ubiquitin would clash with the HUWE1 dimer interface, if it were positioned in the same way as seen for NEDD4-family enzymes.

*Subsection “The crystal structure of a C-terminal region of HUWE1 reveals an asymmetric dimer”, last paragraph: "[…] would thus, in principle, be free to swing […]". The authors should illustrate this by overlaying the structure of molecule A with the other structures of the HUWE1 HECT domain. The speculation that the C-lobe is free to move should be supported by data. Otherwise the mobility of the C-lobe is possible, but not founded on data.*

In response to this comment, we have now included an additional figure (Figure 2—figure supplement 4), superposing molecule A with catalytically relevant states of NEDD4-subfamily enzymes to show that these conformations would, in principle, not clash with the dimer interface of HUWE1.

As noted above, however, we have clarified the text to state explicitly that the ensemble of HUWE1 molecules is most likely auto-inhibited in solution overall, due to the transient nature of the dimer (subsection “The disruption of the dimerization interface enhances the activity of HUWE1 in vitro and in cells”, first paragraph; Discussion, third paragraph). We comment on the putative conformational space of the C-lobe of molecule A, independent of the lattice environment, solely as part of a comprehensive description of the crystal structure per se(subsection “The crystal structure of a C-terminal region of HUWE1 reveals an asymmetric dimer”, last paragraph) and state in the following section, when discussing solution-based studies, that the dimer is likely not asymmetric from a functional perspective in solution (subsection “The disruption of the dimerization interface enhances the activity of HUWE1 in vitro and in cells”, first paragraph; Discussion, third paragraph).

*5) There are some gaps in the p14ARF experiments that may require additional experiments. It is stated as "data not shown" that a region of p14ARF was identified that bound to HUWE1 (residues 43-75). It is further shown that a peptide corresponding to this region stabilized the dimeric form of a dimerization-competent HUWE1 construct, but had no effect on the isolated HUWE1. This implies that p14ARF, as well as the p14ARF peptide, binds to HUWE1 outside of the HECT domain, but this does not appear to have ever been tested, either here or in the literature. Given the reagents that the authors have in-hand, it seems that it would be very straight-forward to determine the binding site on HUWE1 for p14ARF.*

We thank the reviewers for this suggestion that we have followed up on with fluorescence polarization experiments using a fluorophor-labeled p14ARF-derived peptide and a series of HUWE1 constructs of different lengths (Table 3; Figure 10). These experiments show, intriguingly, that the p14ARF-derived peptide interacts most tightly with the activation segment of HUWE1. The corresponding dissociation constants reside in the low μM range (3 to 7 µM). Residual binding is detected for constructs lacking the activation segment (230 – 340 µM) (subsection “The HUWE1-inhibitor p14ARF interacts with the activation segment and promotes oligomerization of HUWE1”, second paragraph).

Taking into account the results of our SEC studies (Figure 10) and activity assays (Figure 10—figure supplement 1), we propose that p14ARF has a dual role in regulating HUWE1: By interacting with the activation segment, p14ARF may release this segment from the intramolecular engagement with the dimerization region, thus promoting dimer formation and shifting the conformational equilibrium towards the inhibited state. Additionally, p14ARF can inhibit the activity of HUWE1 through interactions with the HECT domain, at least in the context of truncated constructs lacking the activation segment. These data and conclusions have been added to the text (Discussion).

*Finally, as stated in the Discussion, tumor-associated mutations have been identified in HUWE1 that fall within the dimerization domain. The prediction is that these mutations might result in hyper-activation of HUWE1 by inhibiting formation of the dimer. This result would add significantly to the impact of the current paper and it seems that this could be tested with relatively little additional effort.*

In response to this comment and to further interrogate the effects of cancer-associated mutations in HUWE1, we purified the mutated variant, H3962D, of HUWE1 (3951-4374) and subjected it to SEC-MALS analysis (Figure 4). Consistent with the central position of His 3962 at the dimer interface (Figure 4), we find that the dimerization capacity of the mutated protein variant is impaired. The effect is equally strong as the one seen for the mutated variants, I3969A and F3982A (Figure 4), that we had selected based on the structure (subsection “Key contacts in the crystallographic dimer interface mediate the dimerization of HUWE1 in vitro and its self-association in cells”, sixth paragraph).

Since Ile 3969 happens to be a tumor-associated mutation site, as well, our study now provides evidence that two tumor-associated mutation sites in the dimerization region are, indeed, critical for dimer formation. Furthermore, we show that the mutated variants have enhanced activities in ubiquitination assays (Figure 6; subsection “The disruption of the dimerization interface enhances the activity of HUWE1 in 335 vitro and in cells”, first paragraph). We, therefore, speculate that there may be selective pressure, at least in some human tumors, to enhance the activity of HUWE1 (Discussion, eleventh paragraph).